# Native doublet microtubules from *Tetrahymena thermophila* reveal the importance of outer junction proteins

Shintaroh Kubo[1,2,5], Corbin S. Black [1,2,5], Ewa Joachimiak[3,5], Shun Kai Yang[1,2], Thibault Legal[1,2], Katya Peri[1,2], Ahmad Abdelzaher Zaki Khalifa[1,2], Avrin Ghanaeian[1,2], Caitlyn L. McCafferty[4], Melissa Valente-Paterno[1,2], Chelsea De Bellis[1], Phuong M. Huynh[1], Zhe Fan[1], Edward M. Marcotte [4], Dorota Wloga[3] ✉ & Khanh Huy Bui [1,2] ✉

Cilia are ubiquitous eukaryotic organelles responsible for cellular motility and sensory functions. The ciliary axoneme is a microtubule-based cytoskeleton consisting of two central singlets and nine outer doublet microtubules. Cryo-electron microscopy-based studies have revealed a complex network inside the lumen of both tubules composed of microtubule-inner proteins (MIPs). However, the functions of most MIPs remain unknown. Here, we present single-particle cryo-EM-based analyses of the *Tetrahymena thermophila* native doublet microtubule and identify 42 MIPs. These data shed light on the evolutionarily conserved and diversified roles of MIPs. In addition, we identified MIPs potentially responsible for the assembly and stability of the doublet outer junction. Knockout of the evolutionarily conserved outer junction component CFAP77 moderately diminishes *Tetrahymena* swimming speed and beat frequency, indicating the important role of CFAP77 and outer junction stability in cilia beating generation and/or regulation.

Cilia are hair-like structures protruding from the cell surface, conserved from protists to humans. Immotile sensory cilia play an essential role in sensing and transducing external signals such as sonic hedgehog from the surrounding environment to the cell. Coordinated beating of motile cilia enables transport of cells or fluids along the surface of ciliated cells, and thus cilia are key factors in clearing mucus out of the respiratory tract or circulating cerebrospinal fluid[1]. The core structure of motile cilia, the axoneme, consists of two central singlet microtubules and nine peripherally positioned doublet microtubules (DMT)[2,3]. The assembly and stability of the DMTs are indispensable for the ciliary function as they serve as a scaffold for numerous ciliary complexes, the force transducers for ciliary bending[4], and tracks for intraflagellar transport[5].

The inner (luminal) wall of the DMT is supported by an intricate network of microtubule inner proteins (MIPs)[6–8]. Using single-particle cryo-electron microscopy (cryo-EM), the identities of over 30 MIPs have been revealed in the ciliate *Tetrahymena thermophila*[4,9,10], the green algae *Chlamydomonas reinhardtii*[11], and bovine respiratory cilia[12].

At least half of the MIPs identified in those studies are conserved, including the inner junction proteins FAP20, PACRG[9,13,14], FAP45, FAP52[15], and the protofilament (PF) ribbon proteins, RIB43a and RIB72[16,17]. Interestingly, some MIPs show poor conservation. For example, the tektin bundles of the PF ribbon were identified in bovine respiratory cilia, but not in *Tetrahymena* or *Chlamydomonas*[12].

There are few studies exploring the functions of MIPs. RIB72/ EFHC1, an EF-hand protein, is implicated in juvenile epilepsy

[1]Department of Anatomy and Cell Biology, McGill University, Montreal, QC, Canada. [2]Centre de Recherche en Biologie Structurale, McGill University, Montreal, QC, Canada. [3]Laboratory of Cytoskeleton and Cilia Biology, Nencki Institute of Experimental Biology of Polish Academy of Sciences, 3 Pasteur Street, 02-093 Warsaw, Poland. [4]Department of Molecular Biosciences, Center for Systems and Synthetic Biology, University of Texas, Austin, TX 78712, USA. [5]These authors contributed equally: Shintaroh Kubo, Corbin S. Black, Ewa Joachimiak. ✉e-mail: d.wloga@nencki.edu.pl; huy.bui@mcgill.ca

disorder[18,19] and is important for *Tetrahymena* cell motility[20,21]. FAP45 and FAP52 play a role in anchoring the B-tubule to the A-tubule wall; a lack of FAP45/FAP52 leads to B-tubule instability[15]. Pierce1 and 2, the orthologs of *Chlamydomonas* FAP182[11], are important for outer dynein arm assembly in mammals and zebrafish[12].

Until now, the identity of MIPs at the outer junction of the DMT remains unknown. The outer junction is thought to be the site of B-tubule assembly[22]. Disruption of the outer junction leads to DMT damage and consequently dysfunctional cilia. At the outer junction, α- and β-tubulins from PF B1 form a non-canonical interaction with tubulins of A-tubule[8]. Cleavage of the C-terminal tails of tubulins by subtilisin enables B-tubule formation in vitro[22]. On the other hand, C-terminal tails of tubulins are essential for ciliary function in vivo[23,24]. Therefore, there must be a mechanism that suppresses the C-terminal tails of tubulins of the A-tubule enabling the formation of the B-tubule.

We hypothesized that MIPs belong to one of two groups: evolutionarily conserved "core" MIPs, which are ubiquitous in cilia function; or species or lineage-specific MIPs that contribute to the stability required for some specific movement of cilia and flagella in some species or tissues. Here, we used an integrated approach combining single-particle cryo-EM, mass spectrometry, and artificial intelligence to model MIPs in the DMT lumen of the ciliate *Tetrahymena thermophila*. We showed that nearly half of the MIPs identified in *Tetrahymena* are evolutionarily conserved. In contrast to DMTs in *Chlamydomonas reinhardtii* flagella[11] and bovine respiratory cilia[12] where the MIP network adopts a 48-nm periodicity, the *Tetrahymena* MIP network repeats every 96-nm. In addition, we observed filaments on the outer surface of the DMT, which have implications for intraflagellar transport. Finally, we used a combination of genetics and microscopy techniques to show that CFAP77, the conserved MIP found at the outer junction of the DMT, is important for DMT stability and ciliary motility.

## Results

### *Tetrahymena* DMT consists of conserved and non-conserved MIPs

We optimized the purification of DMTs from isolated *Tetrahymena* cilia to preserve all the associated structures ("Methods"). This approach was successfully used previously to obtain the outer dynein arm-bound DMT structure[25]. The single-particle cryo-EM analyses of the DMT from wild-type (WT) and *K4OR* cells allowed us to obtain a 48-nm repeat structure with a global resolution of 4.1 and 3.7 Å (Fig. 1A and Supplementary Fig. 1A). Focused refinement of overlapping regions that collectively encompass the WT DMT resulted in maps with a resolution range of 3.6 to 3.9 Å (Supplementary Fig. 1B, Table 1, and "Methods"). In addition, we obtained the focused refined DMT structures at 3.2–3.6 Å resolution of the *K4OR* strain, in which lysine residue 40 of α-tubulin is mutated to arginine to prevent acetylation[25,26]. Since the mutation does not seem to affect the overall MIP decoration, we performed many of our analyses in the *K4OR* maps due to the higher resolution.

To identify and model the MIP network, we devised a strategy that employed: (i) homology modeling of conserved MIPs; (ii) de novo identification and modeling of unknown MIPs using artificial intelligent backbone tracing DeepTracer[27] of densities in our map, artificial intelligent structure prediction ColabFold[28,29] of all proteins present in the proteome of the DMT, and structure similarity search of protein backbones within the predicted structures of the proteome of the DMT; and (iii) automatic determination of sequences of a backbone trace using the *FindMySequence* program[30] ("Methods" and Supplementary Fig. 1C). To validate the protein identities, we used multiple techniques and independently arrived at the same identity (Supplementary Tables 1–3). We used *FindMySequence* to compare the E-value of our candidate and the next best hits (Supplementary Table 4). We also examined side-chain fitting of all protein models and cross-links identified from cross-linking mass spectrometry of the cilia in situ

(Supplementary Fig. 2A, B and Supplementary Table 5)[31]. With such an approach, we could identify and validate almost all *Tetrahymena* MIPs and show that they are forming a weaving network inside the DMT (Table 2 and Supplementary Tables 1–4). Many of those MIP densities were missing from our previously reported structure of salt-treated DMT[4] such as TtPACRGA/B, TtCFAP20, TtOJ3, TtRIB22, and TtRIB27A/B.

Among the 42 identified proteins (Fig. 1B–I, Supplementary Fig. 2, Supplementary Tables 1–3, and Supplementary Movie 1), approximately half are conserved in different species (the "core" MIPs) while the rest are species-specific MIPs. Interestingly, many MIPs represented by a single ortholog in *Chlamydomonas*, and bovine had two or more paralogs (denoted by A, B and C) in *Tetrahymena* (see Supplementary Table 1–3 and Supplementary Fig. 3A–H). Many paralogs have alternating patterns along the DMT; for example, we were able to discern between the paralog densities in our map when we modeled TtRIB72, TtPACRG, TtCFAP182, TtRIB27, and TtFAM166. In some instances, such as TtCFAP106 and TtCFAP77, both paralogs were detected in our ciliary mass spectrometry data, but we could not distinguish their densities using cryo-EM due to sequence similarity. In other cases, certain paralogs were present in significantly lower stoichiometry (Supplementary Table 3) and were therefore not modeled such as TtCFAP52B/C. It is possible that those paralogs co-localize along the entire cilium length or that paralogs differentially localize to proximal or distal ends. Therefore, we modeled the most abundant paralogs (Supplementary Tables 1–3).

FAP52, FAP45, FAP106, PACRG, and FAP20 (Fig. 2A) are core MIPs of the inner junction region in *Tetrahymena*[9], *Chlamydomonas*[9,11], and bovine respiratory cilia[12]. Interestingly, *Tetrahymena* CFAP52 is stabilized by an ancillary protein IJ34 (homolog of CCDC81, this study) while in *Chlamydomonas* and bovine respiratory cilia FAP52 is stabilized by species-specific FAP276[9,11] and EFCAB6[12] respectively. The two *Tetrahymena* paralogs of PACRG, TtPACRGA and TtPACRGB differ in their N-terminal region and are positioned alternately in the inner junction (Supplementary Fig. 3B). Consequently, the *Tetrahymena* inner junction has a 16-nm periodicity, which clearly defines the bimodality of the interdimer distance first reported in *Tetrahymena*[4]. The lack of the N-terminal region in TtPACRGB (Supplementary Fig. 3B) might explain why the inner junction of *Tetrahymena* is less stable upon salt treatment compared to *Chlamydomonas*[9], pointing to the role of the N-terminal region of PACRG in inner junction stabilization. We did not see the unknown MIP densities previously reported in the subtomogram averaged map of the *Tetrahymena* axonemal doublet[10]. Perhaps, it is due to partial decoration of this unknown MIP.

The PF ribbon is the most divergent region in the DMT (Fig. 2B). RIB43a, the core filamentous MIP, is conserved across species and was reported previously to be crucial for the stability of the PF ribbon region[4,17]. In *Tetrahymena*, the PF ribbon region is composed of TtRIB72A, TtRIB72B, TtRIB43A_S, TtRIB43A_L, TtCFAP143, TtRIB27, TtRIB57, TtRIB35, and TtRIB26 (Fig. 2B). However, some of those proteins (TtRIB27, TtRIB57, TtRIB35, and TtRIB26) are not conserved in *Chlamydomonas*[11] and bovine respiratory cilia[12] (Supplementary Tables 1 and 2). Moreover, in bovine respiratory cilia, the PF ribbon is accompanied by tektin bundles (Fig. 2B)[12]. The divergence in the PF ribbon components between species demonstrates that the stability of the PF region is not dependent upon widely conserved proteins. Interestingly, three molecules of RIB57 displayed different conformations in the 48-nm doublet (Supplementary Fig. 3H). This conformational flexibility explains the observed length violation of intra-cross-links found in RIB57 in the *Tetrahymena* cilia when compared with the AlphaFold2 model[31].

We observed similarities and differences in outer junction structures and protein composition between *Tetrahymena*, *Chlamydomonas*, and bovine respiratory cilia (discussed later). To summarize, each DMT region has core components while other MIPs are species-

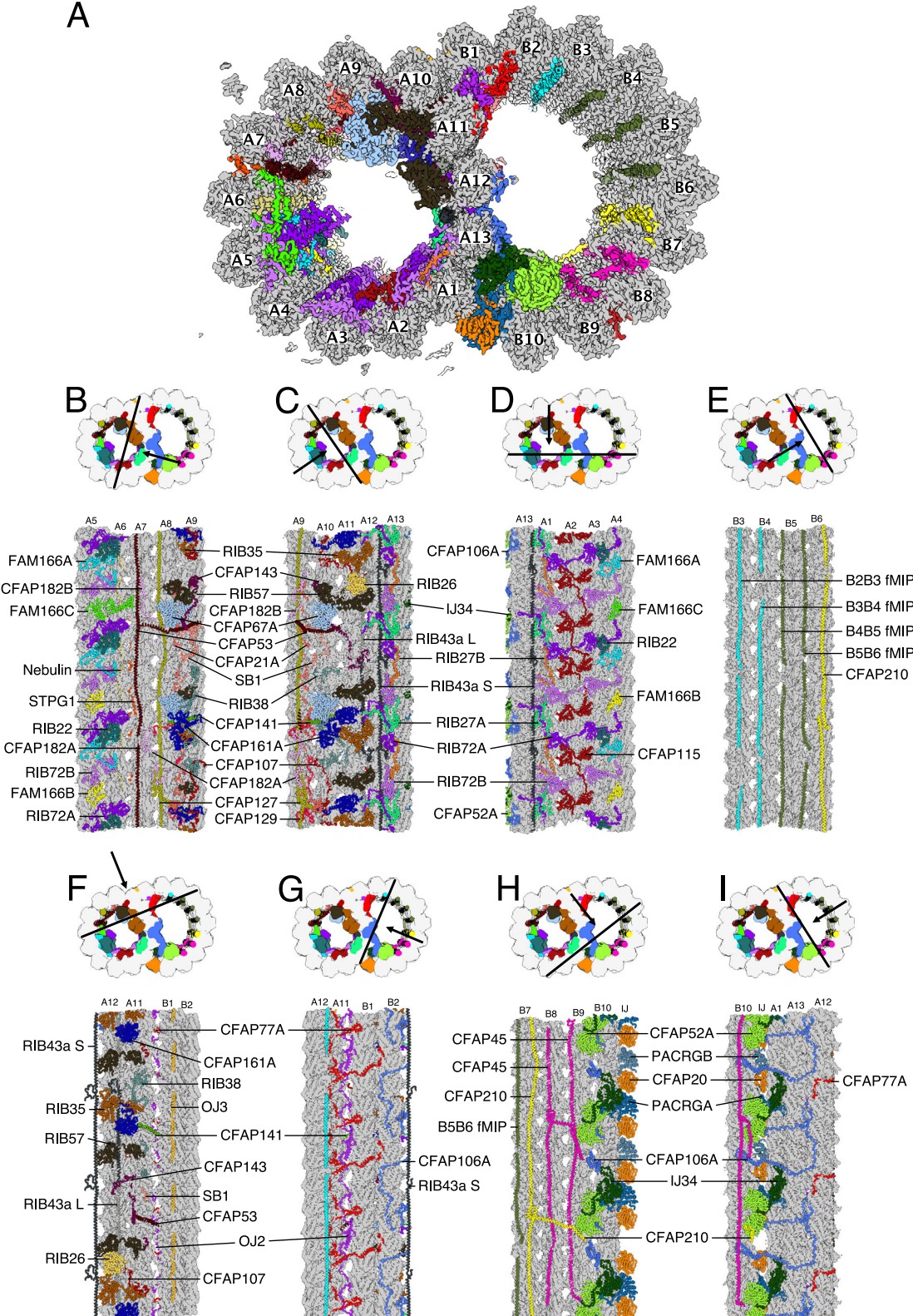

**Fig. 1 | The structure of the native DMT from *Tetrahymena thermophila*. A** A cross-section of the DMT map. Each color denotes an individual MIP. Tubulins are in gray. **B–I** Views of the lumen of the DMT from different angles as indicated by the black arrow. The cutting plane indicated by black lines.

specific. These species-specific proteins possibly function to reinforce core components and contribute to species-specific motility. As such, species-specific MIPs may be essential for cilia stability and waveform regulation.

## MIP distribution in *Tetrahymena* has a 96-nm periodicity

The DMT outer surface has a 96-nm periodicity regulated by the molecular ruler composed of CCDC39 and CCDC40[32]. In *Chlamydomonas*[11] and bovine respiratory cilia[12], the MIPs show a 48-nm

**Table 1 | Cryo-EM data collection and refinement parameters for all datasets used in this study**

| Method | Single particle analysis | | | Subtomogram averaging | |
|---|---|---|---|---|---|
| **Dataset** | **WT (CU428)** | **K4OR** | **96 nm combined** | **WT (CU428)** | **CFAP77AB-KO** |
| Microscope | Titan Krios | Titan Krios | Titan Krios | Titan Krios | Titan Krios |
| Electron Detector | Gatan K3 | Gatan K3 | Gatan G3 | Gatan K3 | Gatan K3 |
| Zero-loss filter (eV) | 30 | 30 | 30 | 20 | 20 |
| Magnification | 64,000 | 64,000 | 64,000 | 42,000 | 42,000 |
| Voltage (keV) | 300 | 300 | 300 | 300 | 300 |
| Electron exposure (e/A$^2$) | 45 | 45 & 73 | 45 & 73 | 160 | 160 |
| Defocus range (μm) | 1.0–3.0 | 1.0–3.0 | 1.0–3.0 | 2.5–3.5 | 2.5-3.5 |
| Pixel size (Å) | 1.37 | 1.37 | 1.37 | 2.12 | 2.12 |
| Tilt range (increment) | – | – | – | −60° to 60° (3°) | −60° to 60° (3°) |
| Tilt scheme | – | – | – | Dose symmetric | Dose symmetric |
| Movies acquired | 18,384 | 25,610 | 43,994 | – | – |
| Particles number | 148,365 | 182,355 | 172,223 | – | – |
| Tilt series acquired | – | – | – | 58 | 20 |
| Subtomograms averaged | – | – | – | 2608 | 1702 |
| Symmetry imposed | C1 | C1 | C1 | C1 | C1 |
| Repeat unit (nm) | 48 | 48 | 96 | 96 | 96 |
| Map resolution | 3.6 to 4.0 | 3.3 to 3.5 | 3.75 | 19 | 22 |

**Table 2 | Refinement statistics of WT and K4OR 48 nm models**

| | **WT (CU428)** | **K4OR** |
|---|---|---|
| Model-to-Map fit, CCmask | 0.8009 | 0.7961 |
| All-atom clashscore | 16.61 | 13.21 |
| **Ramachandran plot** | | |
| Outliers [%] | 0.14 | 0.14 |
| Allowed [%] | 3.70 | 3.60 |
| Favored [%] | 96.16 | 96.26 |
| Rotamer outliers [%] | 0.03 | 0.04 |
| Cbeta deviations [%] | 0.00 | 0.00 |
| Cis-proline [%] | 4.56 | 4.56 |
| Cis-general [%] | 0.01 | 0.01 |
| Twisted proline [%] | 0.17 | 0.18 |
| Twisted general [%] | 0.02 | 0.01 |

periodicity. Interestingly, based on our observations, the *Tetrahymena* DMT has a 96-nm luminal repeat.

*Tetrahymena* TtCFAP115 is almost four times the size of the CrFAP115 (110kD vs. 26kD). The ColabFold[29] prediction of TtCFAP115 shows that it contains four EF-hand pair domains (Interpro IPR011992) while CrFAP115 has only one EF-hand pair domain and repeats every 8 nm. We expected that TtCFAP115 should have a 32-nm periodicity. To accurately position TtCFAP115, we modeled it in a 96-nm map combining several datasets that were obtained at 3.7 Å resolution for *Tetrahymena* WT CU428 strain, and K40 acetylation-deficient strains with α-tubulin point mutation *K4OR*[33]. As expected, TtCFAP115 shows a clear 32-nm repeat (Supplementary Fig. 3I). Since TtCFAP115 repeats at 32-nm while other MIPs such as TtRIB43a repeats at 48-nm, the *Tetrahymena* DMT must have a 96-nm luminal repeat consisting of three TtCFAP115 molecules per periodicity (Fig. 3A, B).

We observed that a single TtCFAP115 molecule interacts with two TtRIB72A and two TtRIB72B molecules (Fig. 3A). To support the prediction of interactions between TtRIB72 and TtCFAP115, we analyzed ciliary proteomes of *RIB72B-KO* and *RIB72A/B-KO Tetrahymena* knockout mutants[21] by mass spectrometry (Fig. 3C and Supplementary Table 6). The top six proteins missing from the *RIB27A/B-KO* mutant are indeed MIPs, highlighting the accuracy of our proteomic analysis.

Interestingly, TtCFAP115 was only missing from *RIB72A/B* double knockout mutant cilia. Our proteomic analysis showed that FAM166C, FAM166B, TtRIB22, and TtRIB27A are also missing only from the *RIB72A/B* double knockout mutant. In contrast, TtRIB27B is significantly reduced in *RIB72B-KO* and completely missing from *RIB72A/B-KO* cilia. Thus, similar to TtCFAP115, stable docking of FAM166B, FAM166C, TtRIB22 and TtRIB27A depends on TtRIB72A but not TtRIB72B (Figs. 1D and 3A), which reveals the role of TtRIB72A and TtRIB72B in the DMT assembly.

Currently no cryo-EM maps of the DMT have shown interactions that explain how the 48-nm repeat of MIPs and the 96-nm repeat of the outside proteins are established by the molecular ruler. Together with existing studies on MIPs[11] and molecular rulers[32], our findings on the 96-nm repeat organization of *Tetrahymena* MIPs suggest that the periodicities of the inside and outside proteins/complexes are self-regulated, probably through head-to-tail interactions of MIPs. Our finding showing that TtCFAP115 interacts with TtRIB72A/B is echoed by recent proteomics and low-resolution tomography studies[10,20]. Moreover, those studies demonstrated that knockout of TtCFAP115 leads to prolonged power stroke and decreased swimming speed[10,20]. Those findings further illustrate the important roles of TtCFAP115 in *Tetrahymena* cilia which are probably a by-product of DMT stabilization. Our structural analysis of TtCFAP115 highlights the importance of high-resolution reconstructions and the power of artificial intelligence prediction in the accurate validation of protein interactions. This strategy allowed us to structurally verify all MIPs that were lost in the mass spectrometry data of the *RIB72A/B-KO* mutants, namely FAM166B, FAM166C, TtRIB22, TtRIB27A, and TtRIB27B (Supplementary Table 6). Such validation is important as mass spectrometry studies often reveal a broad list of missing proteins, many of which are unspecific.

**The outer surface of the intact DMT is associated with many filamentous proteins**

We observed numerous filamentous structures binding to the outer wedge between two PFs on the native DMT outer surface, associated with PFs B1 to B5, and A8 to A10 (Fig. 4A, B). These PFs were shown to serve as tracks for anterograde and retrograde intraflagellar transports (Supplementary Fig. 4A−D)[5].

The low resolution of the outer surface filament suggests that they are either partially decorated or inherently flexible. At this resolution,

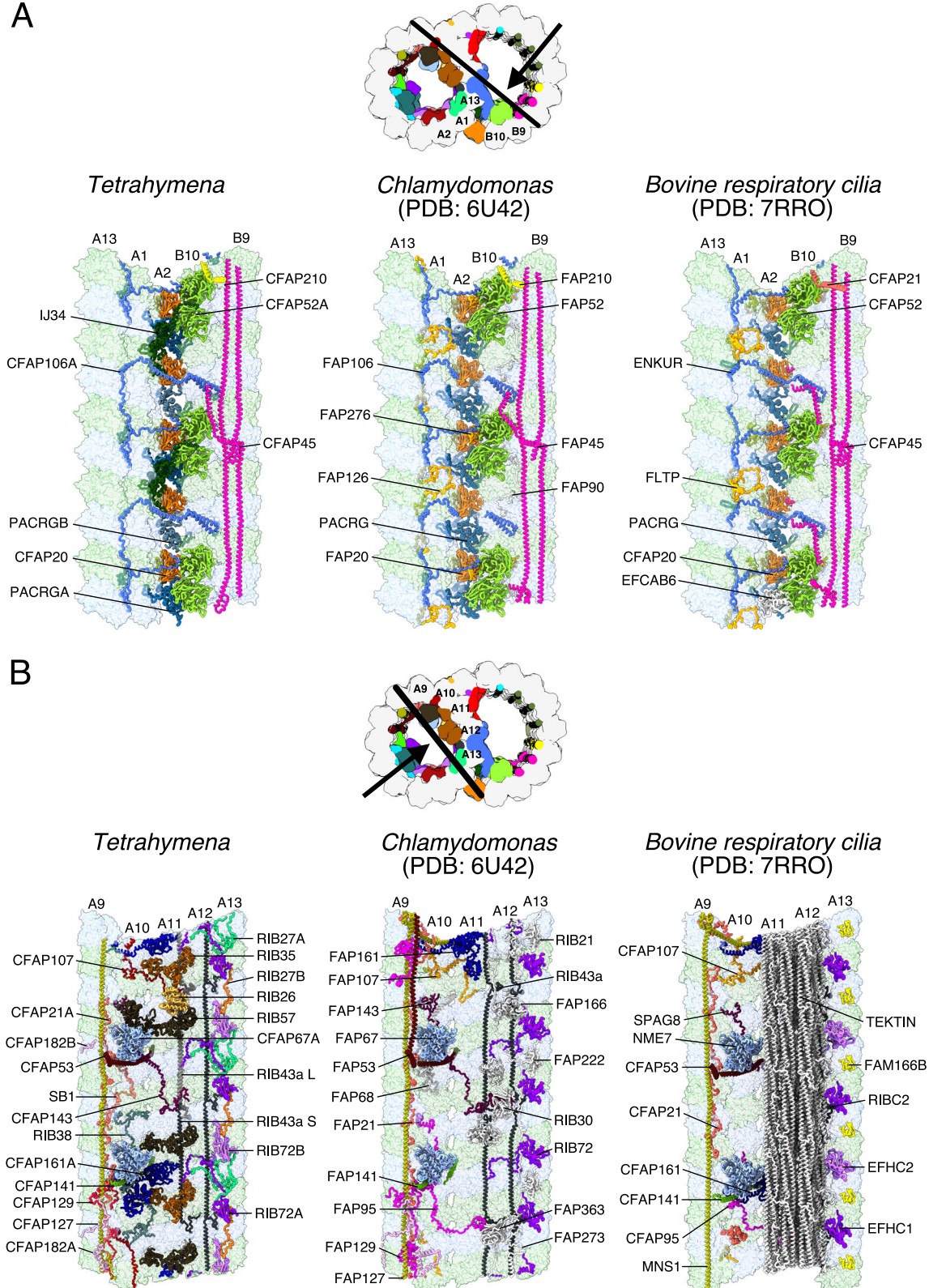

**Fig. 2 | Comparison of the DMT structure from *Tetrahymena*, *Chlamydomonas* and bovine respiratory cilia. A** The inner junction, note that the architecture is well conserved. **B** The PF-ribbon region, note many species-specific MIPs.

the outer surface filaments of PFs A9A10, A10B1 and B1B2 still exhibit a clear 24-nm periodicity (Fig. 4B). Other filaments might have a 48-nm repeat since they appear similarly in the 96-nm map. While we could not identify most of the outer surface, we were able to trace the backbone of the filament between A10 and B1 (OJ3) since it binds tightly to the outer junction wedge and has better resolution. In bovine respiratory cilia and *Chlamydomonas* cilia, a similar density to OJ3 is observed but the identity of the protein remains unknown.

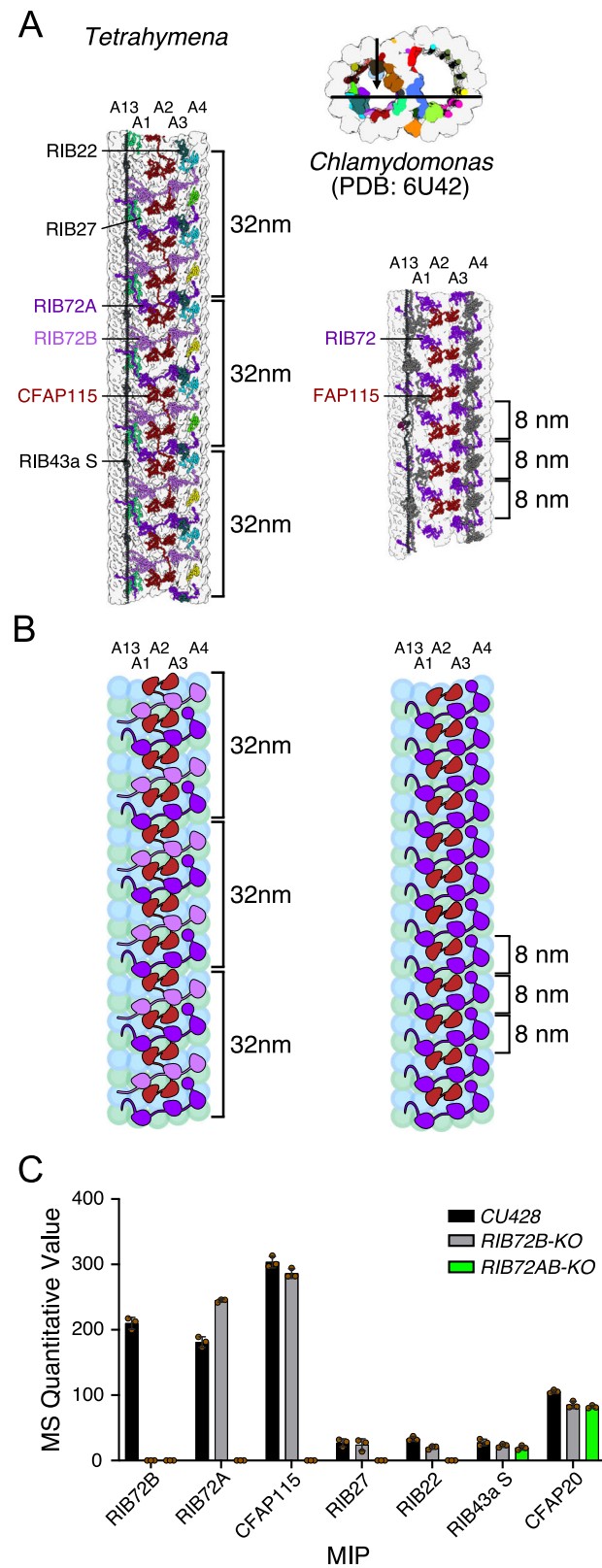

**Fig. 3 | MIPs in *Tetrahymena* exhibit a 96-nm periodicity. A** TtCFAP115 shows 32-nm repeats leading to the true 96-nm periodicity of the MIPs of *Tetrahymena*. The FAP115 in *Chlamydomonas* only repeats with 8-nm periodicity, leading to the normal 48-nm repeat. **B** The cartoon of *Tetrahymena* and *Chlamydomonas* PFs A13-A4 showing the arrangement of FAP115 and RIB72. **C** Quantitative value of mass spectrometry (normal total spectra value) of WT, *RIB72B-KO* and *RIB72A/B-KO* cilia showing that TtCFAP115 is still intact after the knockout of *RIB72B*. *n* = 3 biological replicates for WT and each mutant. Data are presented as mean values +/− standard deviation. Values for each replicate are shown in brown circles. Source data are provided as a Source data file.

filaments. When we overlapped the kinesin and dynein microtubule-binding domains on our DMT structure, we noticed a clear steric clash (Supplementary Fig. 4B−D), similar to the case of MAP7[34]. Thus, likely there is a conformational change upon molecular motor binding to facilitate intraflagellar transport in the cilia. High concentrations of MAP7 seem to inhibit kinesin-1 activity[34]. Therefore, it is reasonable to suggest that the low resolution of the outer surface filament is due to partial decoration and not inhibition of kinesin-2 activity.

In addition, we also observed two proteins binding tightly to the trench of the outer wedge of PF B8B9 and PF A6A7 (Fig. 4A, star labels). Since these two proteins bind tightly to the trench of the wedge, we were able to identify them as sperm tail PG-rich proteins STPG1A and STPG2 (Fig. 4C and Supplementary Fig. 4E, F). Interestingly, while most of STPG1A and STPG2 bind outside the DMT, both weave inside the DMT and contain a luminal domain (Fig. 4C). STPG1A contains five PG-rich motifs, each interspersed by about 40 amino acids (Fig. 4D). By superimposing the five STPG1A PG-rich motifs bound on the B-tubule, we discovered that the PG motif is a microtubule-binding motif and contains six residues with a P-G-P-G-x-Y pattern (Fig. 4E). The ~40 amino acids act as 8-nm spacers allowing each PG motif to bind to consecutive tubulins along the PF. That explains how STPG1A repeats and spans the 48-nm repeat without head-to-tail interactions. In our cilia proteome, there are five PG-rich proteins: STPG1A and its homolog STPG1B; STPG2; and two Outer Dense Fiber 3 Like (ODL3L) proteins (Supplementary Fig. 4E). Except for STPG2, all other proteins contain the P-G-P-G-x-Y motif and ~40 amino acid spacers. Both ODF3L proteins contain 12 P-G-P-G-x-Y motifs, suggesting that they bind to 12 tubulins or 96-nm. Interestingly, we found cross-links from both ODF3L proteins to the outer side of tubulins (Supplementary Table 5). For STPG2 the PG-rich motif has a five-residue pattern, consisting of P-G-P-x-Y, and forms a similar structure bound to microtubules (Supplementary Fig. 4F, G). In short, our structure of STPG1A allows us to conclude that the PG motif is a microtubule-binding motif and that ODF3L proteins are MAPs that bind the outer wedge of the microtubule.

### CFAP77 stabilizes the outer junction

Among the identified MIPs are proteins that support the outer junction, which is believed to be the nucleation site for B-tubule formation. We observed three outer junction proteins: TtCFAP77 and OJ2, inside the B-tubule, and OJ3 in the wedge between A10 and B1 (Fig. 5A, B and Supplementary Movie 2). Bioinformatics analyses and examination of cryo-EM maps from *Chlamydomonas* (EMD-20631) and bovine respiratory cilia (EMD-24664) showed that OJ2 is *Tetrahymena*-specific. In contrast, CFAP77 is conserved in many species including humans and *Chlamydomonas* (Supplementary Fig. 5A, B).

*Tetrahymena* has two paralogs of CFAP77, CFAP77A and CFAP77B (Supplementary Fig. 5A and Supplementary Table 3). The mass spectrometry analyses revealed that CFAP77A is almost three times more abundant in cilia than CFAP77B. Because of the higher expression level, we decided to model CFAP77A.

The detailed analysis of the outer junction architecture reveals two MIPs, TtCFAP77 and OJ2, interacting with four outer junction PFs. TtCFAP77 contacts PFs A10, A11, B1, and B2, while OJ2 contacts PFs A11

The outer surface filaments bind to the DMT in a similar fashion to MAP7[34]. MAP7 is known to activate kinesin-1 even though its α-helical microtubule-binding domain recognizes a site on microtubules that partly overlaps with the kinesin-1 binding site. In contrast, dynein motor activity is not impacted by MAP7 binding to microtubules[35]. In *Tetrahymena*, both intraflagellar transport motors, kinesin-2 and dynein-2, must walk on the DMT decorated by these outer surface

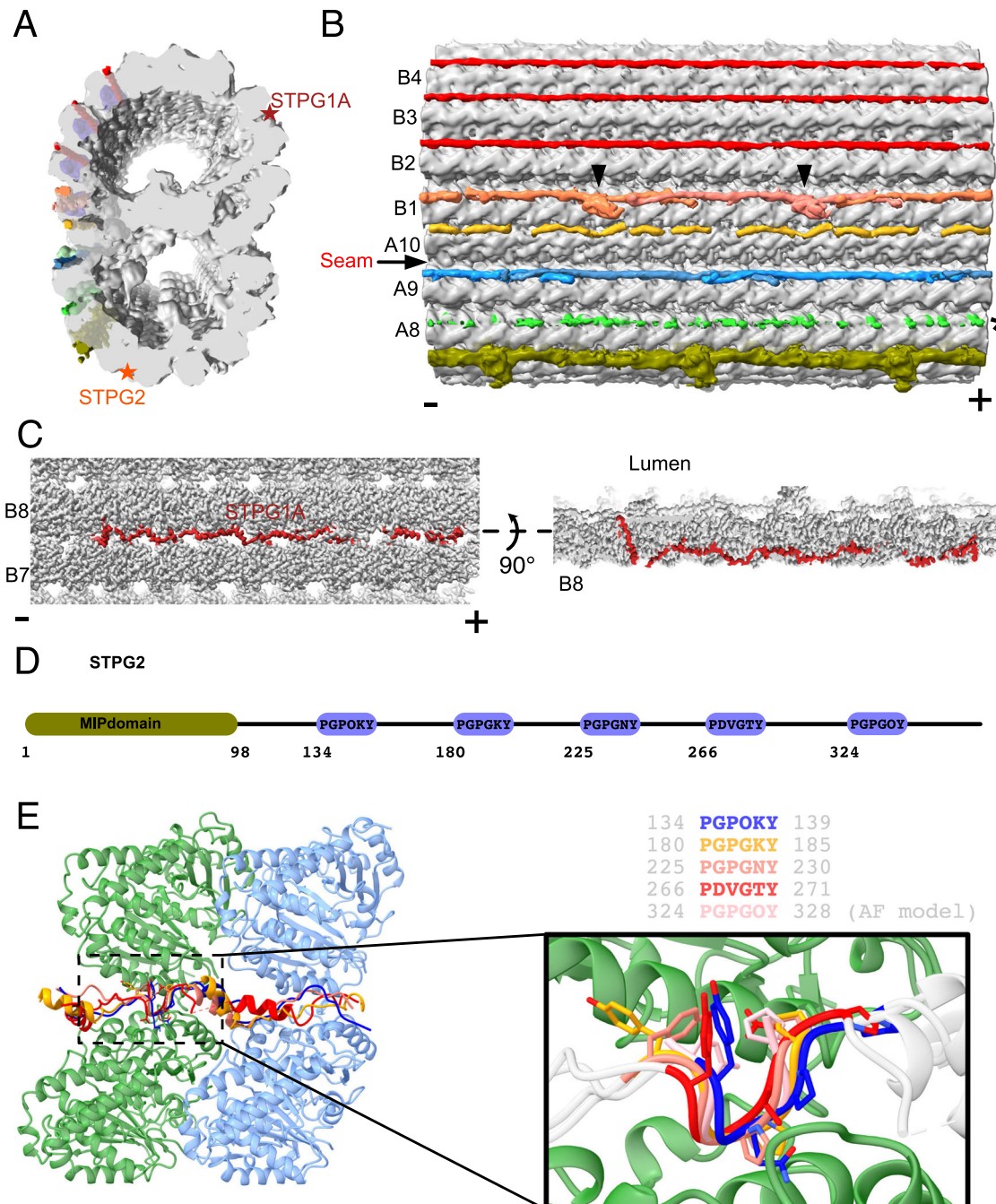

**Fig. 4 | The outer surface filaments on the native DMT. A, B** Cross-sectional (A) and longitudinal (**B**) views of the outer surface filaments on the native *Tetrahymena* DMT. The filaments are bound to adjacent pairs of PFs from A8 to B5. The filaments between PFs B2-B3, B3-B4, and B4-B5 appear similar and have a 48-nm periodicity. In contrast, the filaments between PFs A9-A10, A10-B1, and B1-B2 appear to have a 24-nm periodicity. These filaments have a clear head-to-tail periodic arrangement between PFs A9-A10 and B1-B2. The filament between PFs B1B2 has a globular domain (black arrowheads). The density of the filament between PFs A8-A9 is very weak, probably due to partial decoration. (−) and (+) signs indicate the minus and plus ends of the microtubule respectively. **C** STPG1A is a filamentous protein that is woven between PFs B7-B8, bound to the surface and the lumen. **D** Schematic representation of the protein topology of the filamentous MIP STPG2. **E** The PG-rich repeat motifs of STPG2 are structurally similar.

and B1 (Fig. 5A, B). TtCFAP77 and OJ2 are 29 and 20 kDa acidic proteins, respectively, with short alpha helices interspersed with intrinsically disordered regions (Fig. 5B and Supplementary Fig. 5C, D). OJ3 is a filamentous MAP with four alpha helices broken by short intrinsically disordered regions, positioned between PFs A10 and B1. Unlike the 16 nm repeating unit shared by TtCFAP77 and OJ2 spanning two tubulin dimers, OJ3 repeats every 24 nm, spanning three tubulin dimers (Fig. 5B). Each of these three outer junction proteins acts as a thread that weaves across two or more PFs.

Remarkably, two short helices of TtCFAP77 occupy taxane-binding sites on β-tubulin subunits of PF B2 (Fig. 5C(i and ii)). The helix-turn-helix motif of TtCFAP77 is positioned on the β-tubulin C-terminus of PF A11 (Fig. 5C(iii)). OJ2 is positioned between PFs B1 and A11 and interacts with the taxane-binding site of a single β-tubulin along PF B1 (Fig. 5C(iv)).

We also examine the microtubule-binding loop (M-loop) critical for lateral interactions between PFs B1 and A-tubule and their inter-actions with the outer junction proteins. While the canonical lateral

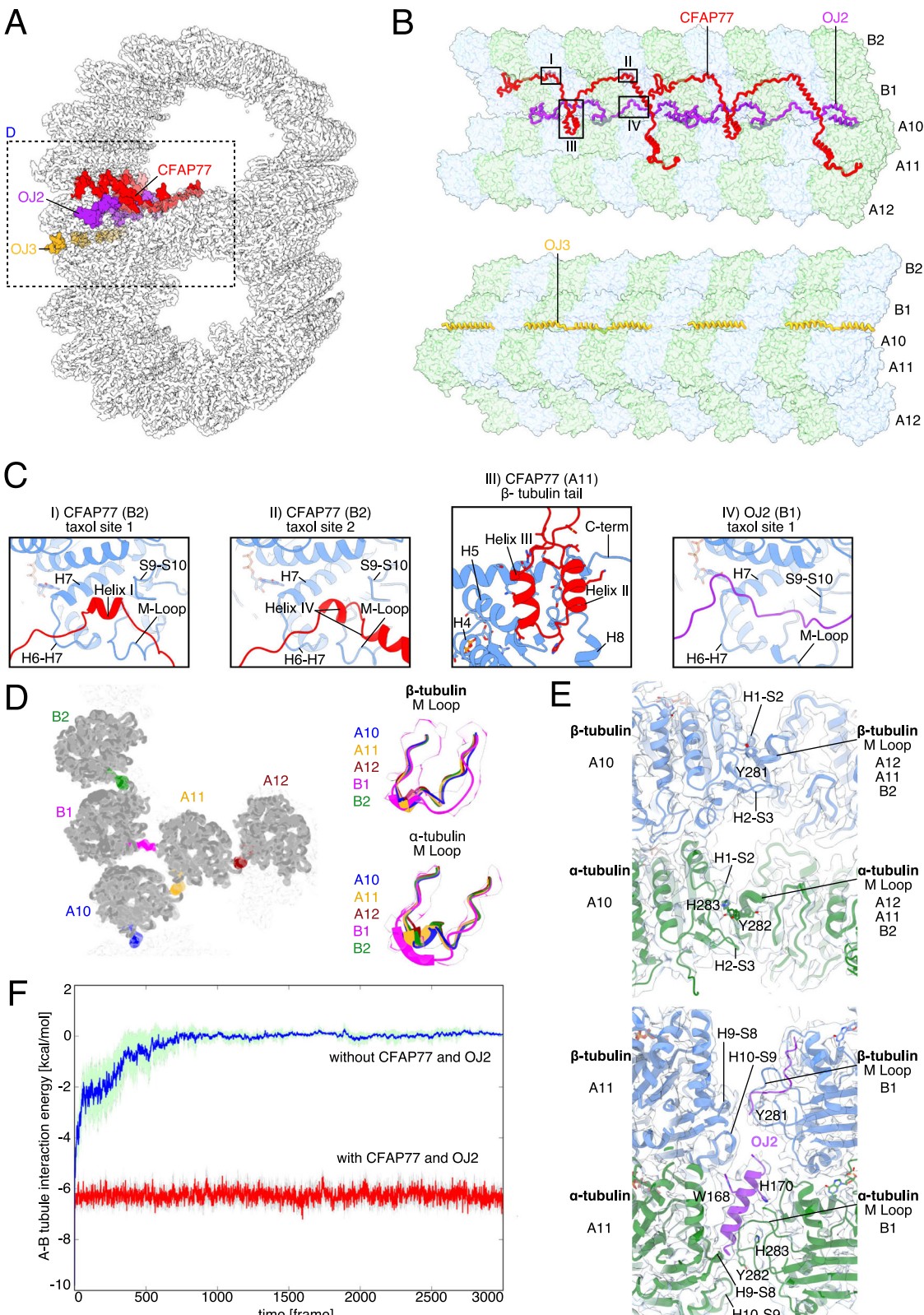

**Fig. 5 | CFAP77 stabilizes the outer junction. A** Cross-sectional view of the *Tetrahymena* outer junction highlighting proteins CFAP77 (red), OJ2 (purple) and OJ3 (yellow). **B** Architecture of the *Tetrahymena* outer junction including proteins CFAP77, OJ2 and OJ3. **C** Helices of CFAP77 occupy taxane-binding pockets of β-tubulin of PF B2. A loop of OJ2 occupies the taxane-binding pocket of β-tubulin of PF B1. The helix-turn-helix motif of CFAP77 is positioned near the C-terminus of β-tubulin of PF A11. **D** Cross-sectional view of the OJ. M-loops of tubulins are colored according to protofilament: B2 (green); B1 (purple); A10 (blue); A11 (gold); A12 (dark red). To the right, those same M Loops are superimposed. **E** The canonical (top) and unique (bottom) lateral interactions between tubulin subunits. PFs A11, A12, and B2 are superimposed and adopt the same overall conformation and lateral interactions (top). The lateral interactions between B1 and A11 are particularly unique in *Tetrahymena* because they involve OJ2 (bottom). **F** Molecular dynamics simulations of the outer junction with and without CFAP77 and OJ2.

interactions between the M-loop and H1-S2 and H2-S3 loops of the adjacent PF[36] are observed throughout the A-tubule and most of the B-tubule, the M-loop of α-tubulin in PF B1 adopts a unique conformation compared to all other M-loops of α-tubulin at the outer junction PFs (Fig. 5D, E). The M-loop of α-tubulin of PF B1 forms hydrogen bonds with both the α-tubulins of A11 and OJ2, providing structural evidence that OJ2 has a stabilizing role in the outer junction (Fig. 5E).

To investigate the significance of TtCFAP77 and OJ2 on the outer junction, we used molecular dynamics simulations to examine the behavior of protofilaments in the presence and absence of TtCFAP77 and OJ2. For the simulation, we used the PFs A10-A12 and B1-B2 as well as TtCFAP77 and OJ2 (see Methods). Our molecular dynamics simulation showed that tubulins from PFs B1-B2 dissociated from PFs A10-A12 in the absence of both TtCFAP77 and OJ2 (Fig. 5F). Molecular dynamics simulations indicate that mainly TtCFAP77 contributes to A- and B-tubule stability because the energy with and without OJ2 is almost the same (Supplementary Fig. 5E). In addition, further analysis shows that B-tubule bounding angle with A-tubule become unstable in the absence of both TtCFAP77 and OJ2 (Supplementary Fig. 5F). Therefore, TtCFAP77 is the key MIP for the stable binding of the B-tubule to the A-tubule.

The helix-turn-helix motif of TtCFAP77 and the wedge position between PFs A11 and B1 of OJ2 might suppress the intrinsically disordered C-terminal tails of β-tubulins during the assembly of the DMT based on its binding position to PF A11 (Supplementary Fig. 5C). Partial DMTs were previously reconstituted in vitro upon the addition of free intact tubulins to subtilisin-treated polymerized microtubules[22]. Formation of partial DMTs was observed only when microtubules lacked β-tubulin C-terminal tails. The pI of TtCFAP77 and OJ2 are 9.49 and 9.56 respectively, while the pI of β-tubulin is 4.79. Thus, under physiological conditions, these proteins carry positive and negative charges, respectively. It is possible that attractive electrostatic interactions between TtCFAP77, OJ2, and the C-terminal tails of β-tubulins are important for the DMT assembly (Supplementary Fig. 5C). In *Tetrahymena*, OJ2 might contribute more to the suppression of the C-terminal tails of β-tubulins due to its positive surface charge and proximity (Supplementary Fig. 5D). The cryo-EM maps of the 48-nm unit of the bovine respiratory cilia and *Chlamydomonas* DMTs have a weak but convincing density for CFAP77 but no equivalent density for OJ2.

## Knockout of CFAP77 moderately reduces *Tetrahymena* cell motility

The structure and molecular dynamics simulation of TtCFAP77 suggested that it plays an important role in the stability of the DMT. Therefore, we investigated how deletion of TtCFAP77 will affect cell swimming, cilia beating, and DMT ultrastructure. Using a germ-line knock-out approach we engineered *Tetrahymena* cells and removed nearly the entire coding region of either *CFAP77A* (*CFAP77A-KO*) or *CFAP77B* (*CFAP77B-KO*), or both genes (*CFAP77A/B-KO*) (Supplementary Fig. 6A, B). Two independent clones were obtained for each mutant. The swimming trajectories of CFAP77 mutants, similar to WT cells, are straight but the cell swimming rate is reduced. Single *CFAP77A-KO* and *CFAP77B*-KO mutants covered approximately 66% and 63%, respectively of the WT distance while double *CFAP77A/B-KO* mutant traveled 58% of the WT distance (Fig. 6A). The analysis of the cilia beating using a high-speed camera revealed that in *CFAP77A/B-KO* mutant ciliary wave form and amplitude were similar to the one observed in the WT cells (Supplementary Fig. 6C) while the ciliary frequency was reduced (Fig. 6B and Supplementary Fig. 6D). Occasionally, in mutants, some asynchronously beating cilia were observed. The cilia length of two clones of *CFAP77A/B-KO* was about 90% that of WT cells, suggesting mild assembly defects (Fig. 6C).

Immunolocalization of TtCFAP77A and TtCFAP77B proteins expressed as fusions with C-terminal 3HA tag under the control of

respective native promoters revealed that TtCFAP77A is present at the proximal end of the cilium while TtCFAP77B localizes along the entire cilium length except for the distal tip (Fig. 6D). However, how TtCFAP77A and TtCFAP77B are targeted for assembly is still unclear.

Using cryo-electron tomography and subtomogram averaging, we obtained structures of the WT and *CFAP77A/B-KO* mutant 96-nm axonemal repeat (Supplementary Fig. 7A, B). The averaged tomographic map of the *CFAP77A/B-KO* mutant axoneme looks similar to WT with no obvious defects. However, when we examined the raw tomograms of the *CFAP77A/B-KO* mutant, we observed gaps in outer junction regions in some of the tomograms (Fig. 6E, F, blue arrows). This suggests that the lack of CFAP77 likely destabilizes the outer junction mildly. We also observed some unknown and non-periodic densities binding to the outer junction from outside in some cilia in the *CFAP77A/B-KO* mutant (Fig. 6E, F, red arrows). Interestingly, the level of tubulin glutamylation in cilia was slightly increased in the *CFAP77A/B-KO* cells compared with the WT cells (Supplementary Fig. 7C–F).

## Discussion

In this work, we used cryo-EM to determine the structure of native DMTs from the ciliate *Tetrahymena thermophila* and a combination of mass spectrometry and artificial intelligence approaches to identify and model proteins inside the DMT. Obtained structures show core MIPs and species-specific MIPs based on conservation with other species.

Since the DMT was obtained without harsh treatment or enzymatic digestion, its structure represents a close-to-native structure. It revealed a difference in periodicity between the outside and inside proteins of the *Tetrahymena* DMT without any clear connection between them. The inside periodicity is a multiple of 16 nm while the outside periodicity is a multiple of 24 nm. In the case of bovine respiratory cilia, it is proposed that Pierce 1 and 2 are the links between the outer 24-nm repeat of the outer dynein arm and the inner 48-nm repeat of MIPs[12]. In *Tetrahymena*, we observed STPG1A and STPG2 bind to the outer wedge and weave into the luminal space of the A- and B-tubule. However, there are many regions of the DMT having no contact between outside and inside proteins. Based on the observation of the outer surface filament, we speculate that the outside periodicity can also be achieved independently through a self-regulated head-to-tail arrangement as suggested for the molecular rulers CCDC39 and CCDD40[32] and MIPs[4,11]. Perhaps initial templates of inside and outside proteins at the base of the cilia initiate the perfect registry between the proteins on the inside and outside surfaces. Interestingly, our structural analysis of the PG-rich motif indicates that ODF3L perhaps span 96-nm in the outer wedge of the microtubules. However, the role of ODF3L in the formation of the 96-nm periodicity of the outside is unknown.

The presence of the MAP7-like outer surface filaments in our structure is very interesting since it might have implications for intraflagellar transport on the DMT. As it was shown that MAP7 has biphasic regulation on kinesin-1 due to competitive binding to microtubules[34], outer surface filaments might impose regulatory effects on dynein and kinesin, partitioning anterograde and retrograde transport tracks. Since there is no evidence that this is a conserved feature in the axoneme[11,12], this might be a species-specific feature for intraflagellar transport in *Tetrahymena*. On the other hand, a sleeve-like density covering the DMT was observed in the *Chlamydomonas* in situ centriole structure[37], which may prevent the continuous contact of intraflagellar transport motors with the DMT. While the mechanisms of intraflagellar transport to overcome the sleeves in *Chlamydomonas* and the outer surface filaments in *Tetrahymena* DMT are supposed to be different, a fundamental question of how the intraflagellar transport trains can accomplish that feat is an intriguing one.

Our study also identifies the conserved protein CFAP77 as an important outer junction protein. In human epithelial cilia CFAP77

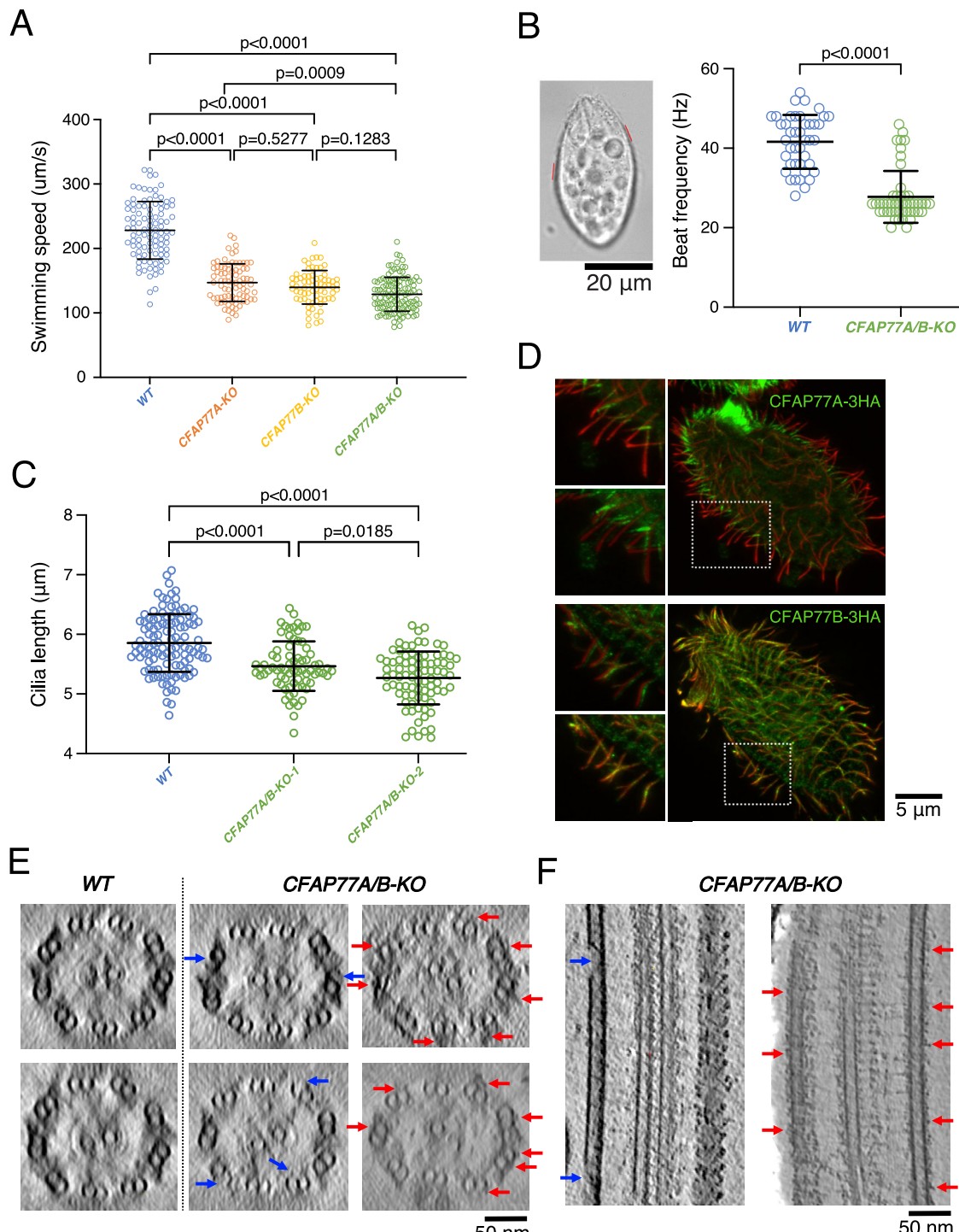

**Fig. 6 | Knockout of *CFAP77A/B* caused mild defects in cilia. A** Knockout of *CFAP77A* or *CFAP77B*, or both led up to 40% swimming speed reduction (*n* = 99 for WT, *n* = 81 for *CFAP77A-KO*, *n* = 70 for *CFAP77B-KO*, *n* = 119 for *CFAP77A/B-KO*). Statistical analyses were done with two-sided Tukey's multiple comparisons tests. **B** *Tetrahymena* cells with marked exemplary positions (red lines) where cilia beat was analyzed in recorded swimming cells. Graphical representation of measurements of cilia beating frequency in WT and *CFAP77A/B-KO* mutants. (*n* = 42 cilia from 12 cells for WT, n = 48 cilia from 12 cells for *CFAP77A/B-KO*). Statistical analyses were done with a two-sided Mann-Whitney comparisons test. **C** Cilia length measurements of WT and *CFAP77A/B-KO* mutants. On average cilia length was: WT = 5.85 μm (number of measured cilia, *n* = 115), *CFAP77A/B-KO* clone 1 = 5.46 μm

(*n* = 74), *CFAP77A/B-KO* clone 2 = 5.27 μm (*n* = 83). Student's *t* test WT/KO is 2E−08 and 6,8E−16, respectively. Data are presented as mean values +/− standard deviation in **A**–**C**. Source data are provided as a Source data file for **A**–**C**. **D** Differential localization of CFAP77A and CFAP77B in *CFAP77A-3HA* and *CFAP77B-3HA* knock-in mutants. Green−anti-HA; red−poly glycylated tubulin. **E** The tomographic cross-sections of WT and *CFAP77A/B-KO* mutants showing occasional damage in the outer junction of *CFAP77A/B-KO* mutants (blue arrows) and unknown densities near the outer junction region (red arrows). **F** Longitudinal sections from *CFAP77A/B-KO* tomogram showing outer junction damage (blue arrows) and unknown densities (red arrows). For (**D**–**F**), the experiments were done once.

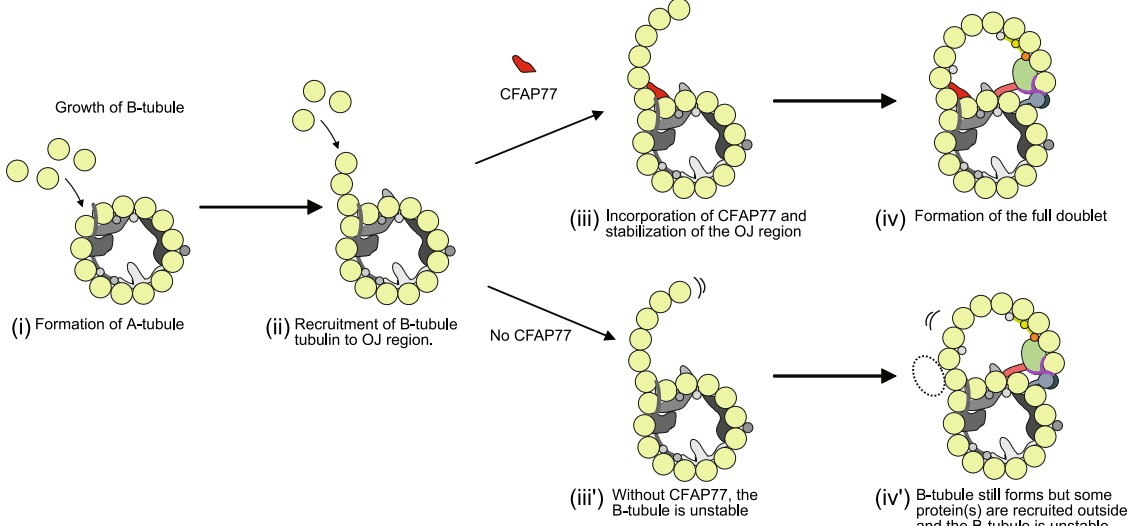

**Fig. 7 | A model of the role of CFAP77 on the formation of the B-tubule.** The A-tubule is formed (i), then free tubulins bind to form the hook at the outer junction (OJ) region (ii). With CFAP77, the newly formed B-tubule hook is stabilized (iii), leading to the final DMT formation (iv). Without CFAP77, the B-tubule is not properly stabilized (iii') and there are additional protein(s) on the outer surface (iv'). The presence of additional protein(s) may or may not be related to elevated levels of tubulin polyglutamylation detected in *CFAP77A/B-KO* cilia.

localizes along the entire cilia length but is most abundant in their middle part[38]. Expression of human CFAP77 increases during ciliated epithelial cell differentiation[38]. In buffalo sperm, CFAP77 is both phosphorylated and ubiquitinated[39]. Hypomethylation of the CFAP77 gene has also been linked to opioid dependence[40]. It is still unclear how to link post-translation modifications of CFAP77 with its function in cilia.

Based on molecular dynamic simulation, knockout phenotype, and mass spectroscopy data we suggest that CFAP77 is important for the swimming of *Tetrahymena* probably through its stabilizing role. Interestingly, the phenotype of *CFAP77A/B-KO* is very similar to *RIB72A/B-KO*[21] and *CFAP115-KO* mutants[20] with lower ciliary beating frequency and coordination of beating between cilia and excessive curve or kink of some cilia. It is possible that MIPs are required for stabilization of the DMT as the force transducer for proper ciliary beating. Lacking certain MIPs reduces the tensile strength of the DMT, which leads to unexpected curves or kinks during ciliary bending and hence affects the beating coordination. Since the force during ciliary bending is stronger at the base, stability of the axoneme base is likely more important for ciliary function. Coincidently, more TtCFAP77A is present in the proximal half of the cilium (Fig. 6C); possibly the B-tubule is more stable in that region.

While TtCFAP77 structure and position suggest that it might be important for the suppression of the C-terminal tails of β-tubulins, our study shows that TtCFAP77 is not vital for the assembly of the DMT as the cilia still form with only occasional damages (Fig. 7). The mild phenotype might be because *Tetrahymena* has the species-specific protein OJ2, which might have a redundant role in ciliary assembly. Thus, the phenotype of a *CFAP77-KO* might be more severe in the case of other species and a potential ciliopathy gene in humans. Based on the increase in polyglutamylation in the *CFAP77A/B-KO* mutants and the presence of proteins outside the DMT, we speculate that the destabilization of the DMT might lead to a compensatory effect in post-translational modifications. Indeed, the increase in polyglutamylation of the DMT might compensate for the de-stabilization from the loss of CFAP77.

Interestingly, CFAP77 was not found in the human primary cilia proteome[41]. Moreover, there is no orthologous protein encoded by the *C. elegans* genome. Yet the DMT exists in the human primary cilia and worm sensory cilia albeit in shorter segments. Therefore, it is likely that CFAP77 contributes more to the stability than the assembly of the outer junction. It was recently shown that Azami Green-fused Tau-derived peptide added to the microtubules induces formation of the more complex structures, including DMT[42] similar to the doublet reconstitution of C-terminal tail truncated tubulins[22]. Therefore, it's possible that other proteins in primary cilia bind to the outer surface of microtubules and interact with or suppress the C-terminal tails of tubulins.

In conclusion, using cryo-EM and improved cilia preparation methods, we have visualized the structure of the native DMT from *Tetrahymena thermophila* and identified 42 MIPs. Moreover, we have shown that an evolutionarily conserved protein, CFAP77, is an outer junction protein and its knockout affects cilia motility (Fig. 7). Our findings contribute to our understanding of MIP function but leave many questions regarding the identity of outer surface proteins and their implication in intraflagellar transport.

## Methods
### Cell culture
All *Tetrahymena* strains (*CU428* WT, *K4OR, CFAP77A-3HA, CFAP77B-3HA CFAP77A-KO, CFAP77B-KO, CFAP77A/B-KO, RIB72B-KO*, and *RIB72A/B-KO* mutants) used in this study were grown in 1 L of SPP media[43] in a shaker incubator at 30 °C and 120 rpm. Cultures were harvested at an $OD_{600}$ of approximately 0.7[44].

### Cilia isolation using dibucaine treatment
*Tetrahymena* cells were harvested and deciliated[44]. Briefly, a 2 L cell culture was harvested by centrifugation at $2000 \times g$ for 10 min at 23 °C. Whole cells were resuspended in fresh SPP and adjusted to a total volume of 24 mL. Dibucaine (25 mg in 1 mL SPP) was added, and the culture was gently swirled for 45 s. To stop the dibucaine treatment, 75 mL of ice-cold SPP supplemented with 1 mM EGTA was added and the dibucaine-treated cultures were centrifuged for 15 min at $2000 \times g$ and 4 °C. The supernatant (cilia) was collected and centrifuged at $30,000 \times g$ for 45 min at 4 °C. The pellet (cilia) was gently washed and resuspended in Cilia Wash Buffer (50 mM HEPES at pH 7.4, 3 mM $MgSO_4$, 0.1 mM EGTA, 1 mM DTT, 250 mM sucrose). The resuspended cilia were flash-frozen with liquid nitrogen and stored at −80 °C.

## Purification of the intact axoneme

The frozen cilia were thawed on ice and then centrifuged for 10 min at 8000 × g and 4 °C. The pellet (cilia) was washed and resuspended in Cilia Final Buffer (50 mM HEPES at pH 7.4, 3 mM MgSO$_4$, 0.1 mM EGTA, 1 mM DTT, 0.5% trehalose). NP-40 Alternative (Millipore Sigma #492016) was added to a final concentration of 1.5% for 45 min on ice and then the de-membranated axonemes were centrifuged. Intact axonemes (membranes removed with detergent) can be used for tomography or further DMT preparation (see below). For tomography, the pelleted intact axonemes were resuspended in Cilia Final Buffer to a final concentration of 3.8 mg/mL. Cross-linking was done with glutaraldehyde at a final concentration of 0.015%. Bradford reagent (Bio-rad #5000201) was used to measure the total protein concentration.

## Purification of DMTs

The intact axoneme pellet (see above) was resuspended with Cilia Final Buffer and ADP was added to a final concentration of 0.3 mM. The sample was incubated at room temperature for 10 min. ATP was added to a final concentration of 1 mM and the sample was again incubated at room temperature for 10 min. DMT samples were adjusted to 2.2 mg/mL using Cilia Final Buffer.

## Cryo-EM sample preparation

C-Flat Holey thick carbon grids (Electron Microscopy Services #CFT312-100) were treated with chloroform overnight. For single-particle analysis, DMT sample (4 µL) was applied to the chloroform-treated, negatively glow-discharged (10 mA, 15 s) grids inside the Vitrobot Mk IV (Thermo Fisher) chamber. The sample was incubated on the grid for 15 s at 23 °C and 100% humidity then blotted with force 3 for 5 s then plunge frozen in liquid ethane.

## Cryo-EM data acquisition

A total of 18,384 and 25,610 movies were collected for the *WT* and *K40R* mutant using the Titan Krios 300 keV FEG electron microscope (Thermo Fisher) equipped with direct electron detector K3 Summit (Gatan, Inc.) and the BioQuantum energy filter (Gatan, Inc.) using SerialEM[45]. The movies were collected with a beam shift pattern of four movies per hole and four holes per movement. The final pixel size is 1.370 Å/pixel. Each movie has a total dose of 45 electrons per Å$^2$ over 40 frames. The defocus range was between −1.0 and −3.0 µm at an interval of 0.25 µm.

## Image processing

Motion correction and dose-weighting of the movies were performed using MotionCor2[46] implemented in Relion 3[47], and the contrast transfer function parameters were estimated using Gctf[48]. Micrographs with apparent drift, ice contamination, and bad contrast transfer function estimation were discarded.

The particles were picked using a combination of manual and automatic pickings to speed up this process. The rare top views of the filaments were picked manually using e2helixboxer[49]. The side views of the filaments were picked automatically using Topaz by training with a set of side view manual picks[50]. The Topaz coordinates were converted into the filament coordinates using a customized clustering and line fitting Python scripts based on RANSAC algorithm.

Particles of 512 × 512 pixels were extracted with 8-nm overlapped, binned twice, and pre-aligned using a modified version of the Iterative Helical Real Space Reconstruction script[51] in SPIDER[52] to work with non-helical symmetry. The alignment parameters were then transferred to Frealign for aligning the particles for six iterations in Frealign[53], and then converted into Relion 3.1. Iterative per-particle-defocus refinement and Bayesian polishing were done for the 8 nm particles.

Each particle was subtracted from its tubulin lattice signal and underwent 3D classification into two classes to obtain the 16-nm repeat particles. The 16-nm repeat particles were then subjected to 3D classification into 3 classes to obtain the 48-nm repeat particles. The 48-nm particles were then refined, resulting in a global resolution of 4.0 Å and 3.5 Å for WT and the *K40R* mutant (Table 1).

To improve the local resolution for modeling, we performed focused refinements of different regions by different masks to cover approximately two PFs. Next, the maps were merged into a composite map by Phenix combine_focused_maps tool[54] and then enhanced by DeepEmhancer[55] to improve visualization and interpretability.

To obtain the 96-nm map, we combined WT and K40R particles and performed a 3D classification into two classes with a mask focusing on the outside of the DMT and subsequent refinement of the 96-nm particles. We obtained a 3.75 Å resolution of the combined 96-nm map (Table 1).

## Modeling

Modeling of tubulins was done based on PDB 6U0H. For MIP modeling, two strategies were carried out. For conserved MIPs with available models from *Chlamydomonas reinhardtii* (PDB 6U42), homologous models of *Tetrahymena* proteins were either constructed using Modeller[56] or predicted using ColabFold[29]. The homologous models were fitted into the 48-nm density map of *Tetrahymena* based on the relative location in the *Chlamydomonas* map or by docking using UCSF ChimeraX function fitmap[57]. The final models were then modeled to the density using Coot[58] and real-space refined in Phenix[54] (Table 2). For unknown densities in the *Tetrahymena* map, connected density was first segmented manually using Chimera. The segmented density was then submitted to DeepTracer server[27] to trace the C-alpha backbone. The C-alpha model was then searched for fold similarity in the library of ColabFold predicted models of all proteins in the proteome of *Tetrahymena* cilia using Pymol *cealign* function. The top candidate ColabFold predicted models were then fitted to the full *Tetrahymena* density map for evaluation. Suitable candidates were gone through modeling in Coot and real-space refinement in Phenix (Supplementary Fig. 1C). Alternatively, we searched for the identity of the proteins by using *FindMySequence*[30] to search the C-alpha backbone trace within the database of the cilium proteome (Supplementary Fig. 1C). In certain cases, we used the same density from the *Tetrahymena* map of *K40R* mutant to facilitate modeling since the *K40R* map has a better resolution at 3.5 Å. For validation, we used both *FindMySquence*'s E-value as an indicator for a good identification and in situ cross-linking mass spectrometry of *Tetrahymena* cilia[31] (Supplementary Fig. 1D, Supplementary Fig. 2, Supplementary Table 5). All of the maps and model visualization were done using ChimeraX[57].

## Cryo-ET sample preparation

To preserve the round shape of the axoneme, the axonemes for tomography were cross-linked by glutaraldehyde (final concentration 0.15%) for 40 min on ice and quenched by 1 M HEPES. Axonemes were approximately 3.6 mg/mL when mixed with 10 nm gold beads in a 1:1 ratio for a final axoneme concentration of 1.8 mg/mL. Cross-linked axoneme sample (4 µL) was applied to negatively glow-discharged (10 mA, 15 s) C-Flat Holey thick carbon grids (Electron Microscopy Services #CFT312-100) inside the Vitrobot Mk IV (Thermo Fisher) chamber. The sample was incubated on the grid for 15 s at 23 °C and 100% humidity then blotted with force 0 for 8 s then plunge frozen in liquid ethane.

## Cryo-ET acquisition and reconstruction

Tilt series were collected using the dose-symmetric scheme from −60 to 60 degrees with an increment of 3 degrees. The defocus for each tilt series ranges from −2.5 to −3.5 µm. The total dose for each tilt series is 160 e⁻ per Å$^2$. For each view, a movie of 10–13 frames was collected. The pixel size for the tilt series is 2.12 Å. Tomograms were reconstructed using IMOD[45]. Frame alignment was done with Alignframes[59]. Aligned

tilt series were manually inspected for quality and sufficient fiducial presence. Batch reconstruction was performed in IMOD. The cross-linked axonemes appear similar to non-cross-linked axoneme without any obvious artifacts.

## Subtomogram averaging

Subtomogram averaging of the 4-times binned 96 nm repeating unit of WT and *CFAP77A/B-KO* mutants (2608 and 1702 subtomograms respectively) was done using the "axoneme align"[60]. CTF estimation was done with WARP[61]. Refinement of the 96 nm subtomogram averages was done with the Relion 4.0 pipeline[62]. The resolutions for the 96-nm repeating unit of WT and *CFAP77A/B-KO* DMT are 18 and 21 Å, respectively. For visualization, tomograms were CTF deconvolved and missing wedge corrected using IsoNet[63].

## Setting up for CFAP77 coarse-grained MD simulation

We used A10-12 and B1-2 PFs, OJ2 and TtCFAP77 to model *Tetrahymena* outer junction region. Based on the atomic model of the outer junction region, coarse-grained MD simulation was performed using CafeMol 2.1[64]. Each amino acid was represented as a single bead located at its Cα position in the coarse-grained model. We used the energy functions AICG2+, excluded volume, and electrostatic interaction for predicting dynamics. In the AICG2+, the original reference structure was assumed as the most stable structure, and parameters were modified to represent the interactions in the all-atom reference structure[65]. Three residues (F133, G308, and E401) at the three A10 α-tubulins were anchored for the convenience of the analysis. It is known that the inter α-β dimer interaction is much weaker than intra α-β dimer interaction. Also, inter A-B tubule interaction is weaker than intra A- or B-tubule interaction. To replicate these features in our simulations, we set inter α-β dimer's interacting force to 0.8 times the original value while the intra α-β dimer's interacting force was left as the original value (1.0 times). Also, we set inter A-B tubule interacting force to 0.2 times the original value while that of intra A- or B-tubules PFs' interacting force was set to 0.3 times the original value.

Then, we performed simulations of the outer junction region stability with and without OJ2 and CFAP77 molecules coarse-grained MD, 20 times for each setup. Each MD simulation took $3 \times 10^7$ MD steps. Note, that one MD step roughly corresponds to approximately 1 ps. The MD simulations were conducted by the underdamped Langevin dynamics at 300 K temperature. We set the friction coefficient to 0.02 (CafeMol unit), and default values in CafeMol were used for other parameters.

To calculate angular elasticity between A- and B-tubules (Supplementary Fig. 5F), the two vectors used in the angle calculation were defined by the K401 of the central α-tubulin in A10 as a vector base point, and the center of mass of the central α-tubulin in A12 and B2 as an endpoint of the vector. Since K401 is a residue anchored in space for MD simulation efficiency, it serves as a base point for following the change in the angle between A and B tubules.

## CFAP77 gene knock-ins and knock-outs

To reveal the localization of the CFAP77 protein paralogs, we amplified the entire open reading frame with the addition of MluI and BamHI restriction sites at 5' and 3' ends, respectively, and 3'UTR with the addition of PstI and XhoI restriction sites at 5' and 3' ends, respectively using Phusion HSII High Fidelity Polymerase (Thermo Fisher Scientific Baltics, Lithuania) and primers listed in Supplementary Table 7, and replaced fragments of the *CFAP44* gene in *CFAP44-3HA* plasmid[66]. Such a construct enabled the expression of the C-terminally 3HA tagged CFAP77 paralogs under the control of their own promoters and selection of the positive clones based on the resistance to paromomycin (neo4 resistance cassette[67]). Approximately 10 μg of plasmids were digested with MluI and XhoI to separate a transgene from the plasmid backbone, precipitated onto DNAdel Gold Carrier Particles

(Seashell Technology, La Jolla, CA, USA) according to the manufacturer's instructions and used to biolistically transform CU428 cells. The positive clones were grown in SPP medium with the addition of an increasing concentration of paromomycin (up to 1 mg/mL) and decreasing concentration of CdCl₂ (up to 0.2 μg/mL) to promote the transgene assortment.

To knock out *Tetrahymena CFAP77* genes we employed the germline gene disruption approach[68,69]. Using primers listed in Supplementary Table 7 we amplified fragments of the targeted genes with the addition of selected restriction sites and cloned upstream and downstream of the neo4 resistance cassette[67]. In both loci, nearly the entire coding region was removed. To obtain strain with deletion of both *CFAP77A* and *CFAP77B*, we crossed *CFAP77A* and *CFAP77B* heterokaryons and followed with cell maturation and selection as described[68]. At least two independent clones were obtained for single and double mutants. The loci deletion was confirmed by PCR (primers in Supplementary Table 7).

## Phenotypic and localization studies

Cells swimming rate and cilia beating were analyzed as previously described[70]. For protein localization studies, *Tetrahymena* cells were fixed either by adding equal volume of the mixture of 0.5% NP40 and 2% PFA in PHEM buffer or by permeabilization with 0.25% Triton-X-100 followed by fixation with 4% PFA (final concentrations) on coverslips, air-dried, blocked using 3%BSA/PBS and incubated overnight with anti-HA antibodies (Cell Signaling Technology, Danvers, MA, USA) diluted 1:200, and polyG antibodies[71], 1:2000 at 4 °C. After washing, 3×5 min with PBS, samples were incubated with secondary antibodies (anti-mouse IgG conjugated with Alexa-488, and anti-rabbit IgG conjugated with Alexa-555 (Invitrogen, Eugene, OR, USA) both in concentration of 1:300. After washing, the coverslips were mounted in Fluoromount-G (Southern Biotech, Birmingham, AL, USA) and viewed using Zeiss LSM780 (Carl Zeiss Jena, Germany) or Leica TCS SP8 (Leica Microsystems, Wetzlar, Germany) confocal microscope. The expression of the tagged proteins was verified by Western blot as described before[72].

## Quantification of polyglutamylation

WT cells were incubated for 10 min in medium supplied with India Ink to form ink-filled dark food vacuoles. Next, WT and *CFAP77A/B-KO* cells were fixed side-by-side as described[73] and stained with polyE anti-polyglutamic acid primary antibody (1:2000). WT and mutant cells position next to one another were recorded using a confocal microscope and the intensity of the polyE signal in cilia was measured using ImageJ program. For western blotting, 5 μg of cilia protein per lane were separated on a 10% SDS-PAGE and western blots were done as described[74], with the primary antibodies at the following dilutions: GT335 anti-glutamylated tubulin mAb (1:1000), 12G10 anti-α-tubulin mAb (1:40,000) deposited to the DSHB by Frankel, J. / Nelsen, E.M. (DSHB Hybridoma Product 12G10 anti-alpha-tubulin), polyE antibodies (1:20,000). The intensity of detected bands was measured using ImageJ and presented as a graph showing a ratio of the glutamylated tubulin to α-tubulin.

## Mass spectrometry

Samples prepared for cryo-EM were also analyzed by mass spectrometry. Approximately 25–30 μg proteins were loaded on the SDS-PAGE gel. Electrophoresis was performed, but the run was terminated before the proteins entered the resolving gel. The gel band was cut, reduced with DTT, alkylated with iodoacetic acid and digested with trypsin. Extracted peptides were re-solubilized in 0.1% aqueous formic acid and loaded onto a Thermo Acclaim Pepmap (Thermo, 75μM ID × 2 cm C18 3uM beads) precolumn and then onto an Acclaim Pepmap Easyspray (Thermo, 75μM × 15 cm with 2μM C18 beads) analytical column separation using a Dionex Ultimate 3000 uHPLC at 250 nL/min with a gradient of 2–35% organic (0.1% formic acid in acetonitrile) over

3 h. Peptides were analyzed using a Thermo Orbitrap Fusion mass spectrometer operating at 120,000 resolution (FWHM in MS1) with HCD sequencing (15,000 resolution) at top speed for all peptides with a charge of 2+ or greater. The raw data were converted into *.mgf format (Mascot generic format) for searching using the Mascot 2.6.2 search engine (Matrix Science) against *Tetrahymena thermophila* protein dataset from UniProt.

Mass spectrometry data were analyzed by Scaffold_4.8.4 (Proteome Software Inc.). Proteins with mean values of exclusive unique peptide count of 2 or more in the WT mass spectrometry results were used for analysis. Raw mass spectrometry data were normalized by total spectra. One-way ANOVA was applied to WT, *RIB72A/B-KO* and *RIB72B-KO* mass spectrometry results using biological triplicates. Proteins missing in *RIB72A/B-KO* and *RIB72B-KO* compared to WT were identified in Supplementary Table 6. We used the emPAI score calculation of ciliary proteins of specific sizes and periodicities as complementary information for protein identification. We compared the emPAI scores between salt-treated and non-salt-treated axonemes to filter out proteins that are on the surface of the microtubule, rather than the lumen.

### Reporting summary

Further information on research design is available in the Nature Portfolio Reporting Summary linked to this article.

## Data availability

The data generated in this study are available in the following databases: 8G2Z (48-nm Tetra WT doublet), 8G3D (48-nm Tetra K40R doublet), EMD-29685 (48-nm Tetra WT doublet), EMD-29692 (48-nm Tetra K40R doublet), EMD-29693 (96-nm Tetra combined doublet), EMD-29667 (96-nm Tetra WT doublet subtomogram average), EMD-29666 (96-nm Tetra CFAP77A/B-KO doublet subtomogram average). All data used but not produced in this study are available in the following databases: 6U0H (48-nm Tetra WT doublet), 6U42 (48-nm Chlamydomonas WT doublet), 7RRO (48-nm Bovine WT doublet), EMD-20602 (48-nm Tetra WT doublet), EMD-20631 (48-nm Chlamydomonas WT doublet), EMD-24664 (48-nm Bovine WT doublet). The cilia length measurement, swimming speed, waveform and polyglutamylation quantification data, original gel and blot, and mass spectrometry of WT, *RIB72B-KO, RIB72A/B-KO* cilia generated in this study are provided in the Supplementary Information/Source Data file. Source data are provided with this paper.

## Code availability

The Python code for filament fitting is available on Github.

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

## Acknowledgements

We thank Dr. Kelly Sears, Dr. Mike Strauss, Dr. Kaustuv Basu (Facility for Electron Microscopy Research at McGill University) for helping with data collection and Dr. Muneyoshi Ichikawa for critically reading of the manuscript. We acknowledge the RI-MUHC

Proteomics and Molecular Analysis Platform for assistance with protein analysis. S.K. is supported by JSPS Overseas Research Fellowships. K.H.B. is supported by the grants from Canadian Institutes of Health Research (PJT-156354) and Natural Sciences and Engineering Research Council of Canada (RGPIN-2022-04774). E.M.M. acknowledges support from the Welch Foundation (F1515) and U.S. National Institutes of Health (RO1 HD085901).

## Author contributions

Conceptualization: K.H.B. Methodology: T.L., K.P., A.G., A.A.Z.K., C.D., M.V.-P., Z.F., P.H., C.L.M. Formal analysis: S.K., C.B., E.J., S.K.Y., C.L.M. Investigation: S.K., C.B., E.J., S.K.Y., T.L., K.P., A.G., A.A.Z.K., C.D., M.V.-P., Z.F., P.H., C.L.M., E.M.M. Resources: D.W., K.H.B. Writing—original draft: K.H.B., D.W., S.K., C.B. Writing—review and editing: K.H.B., D.W., S.K., C.B., E.J., K.P.

## Competing interests

The authors declare no competing interests.
