## [Peer Review File · Nature Communications]

REVIEWER COMMENTS

Reviewer #1 (Remarks to the Author):

In this manuscript, Kubo et al. describe a study on the axoneme in *Tetrahymena*. By using cryoEM and single particle analysis on isolated doublet microtubules (DMT), the authors obtained a structure of DMT at 4.1 angstrom. With the aid of protein structure prediction and modeling, this led to atomic models of 38 MIPs found in the map, including 11 MIPs that may not have been previously identified. In addition, the authors described several filamentous structures decorating on the outer surface of DMT, whose molecular identities remain unknown. Finally, the authors characterized the function of two MIPs at outer junction, CFAP77A/B. By analyzing the ciliary beating phenotype of CFAP77A/B knockout and by comparing its structure to wildtype by cryoET, they found the mutant has slower beating rate than the WT. Overall, the techniques the authors had applied are sound. The article is well written, but the organization is a bit disjointed. While the paper appears to focus on the inner junction, there is additional work on other topics added to the manuscript. For instance, analysis of the rib72A/rib72B mutant strain has been published previously and it is unclear what is added in this study. Furthermore, there are several questions concerning the validation of the results and the conclusions presented by the authors.

Major comments.

- 1) The authors claimed to have identified 11 new MIPs that are unique to *Tetrahymena*. However, the list of MIPs presented here are not well rectified with previous studies of the *Tetrahymena* axoneme. There is some discussion of the topic, but it is difficult to easily identify newly identified MIPs versus previously known MIPs. Perhaps the MIPs listed in Tables S1 and S2 could be annotated to identify previous studies in which the given MIP was identified.
- 2) The authors named non-conserved “auxiliary” MIPs, while the authors classified the rest of MIPs that have been found in other species as “core” MIPs. From an evolution point of view, perhaps this is correct, but this is likely not true as these MIPs’ functions concern. It’s likely that the auxiliary MIPs will be equally important for the cilia stability or waveform regulation. Until further characterizing the function of these MIPs, calling them “core” or “auxiliary” could be misleading about function and other names for the groups should be considered.
- 3) The authors estimated the overall resolution of the 48nm long DMT at 4.1 angstrom. Furthermore, by using focused refinement, the local resolution has been improved to 3.6 to 3.9 Å. The authors need to provide a local resolution estimate for validation .
- 4) Concerning identifying and model building of the MIPs, the authors have adapted two parallel approaches and have provided example for each in Figure S1B. In both examples, the MIPs (identities not indicated in Fig S1B) have a globular domain. Perhaps these two represent the best scenario. It is well known that tracing protein backbone and deciding the direction of chain are difficult when the target structure is most coiled-coil or unstructured loop. This is exacerbated when the map resolution is

lower than 4 Å where the most side-chains are invisible. The authors need to provide detailed examples on these MIPs, for example, RIB38, NIP21A and MIP26.

5) Interestingly, the authors have found the FAP115 having a periodicity of 32 nm. This is consistent with the size TtFAP115 that is about 4 times larger than CrFAP115. However, since the 4 EF domains in FAP115 are not identical, how do authors know which map density is attributed to domain 1,2,3,4 respectively? Fig S3A showed subtle difference among 4 EF-hand domains, but no detail has been provided. As authors correctly pointed out that the unique 32nm periodicity of FAP115 implies that entire DMT having a periodicity of 96 nm. Meanwhile, the location of all MIPs in DMT are coherent, it will be critical to understand the position of FAP115 molecular and its domains relative to other MIPs, for example MIPs with 48 nm periodicity. The authors have not provided this information.

6) The authors found several fMIP at the outer surface of DMT. The sites coincide with the binding sites for kinesin or dynein IFT trains. The authors draw comparison of these fMIPs to MAP7 that reportedly has a biphasic regulatory activity for kinesin-1. However, in this case the kinesin-1 activity increases only when the MAP7 concentration is low and when there are many available sites on MT unoccupied by MAP7. Given that the outer surface of DMT is fully occupied by fMIP, the mechanism of IFT trimeric kinesin-2 binding on that surface will likely be different from that of MAP7 decorated MT. The authors' hypothetical analogy is questionable.

7) The authors studied the CFAP77A/B KO by cryoET and subtomogram averaging, and had observed no difference compared to the WT. Fig 6D shows the two averages but lacking any labels that depict their orientations or indicate the colored densities. Is the CFAP77A/B density missing the double KO average? The authors reported the subtomogram averaged at 18 and 21 angstroms for the WT and the mutant. Without reporting their FSCs or the number of subtomogram used, it is difficult to judge the quality of the results. In Fig 6E and F, the authors reported DMT defects in the raw tomograms, have they tried 3D classification to identify these defects in subtomogram average? Finally, 0.15% glutaraldehyde was used for sample preparation. It is well-known this non-specific cross-linking reagent is prone to introduce artefact. This raises concern on the interpretation of the result from cryoET.

8) In "Material and Methods", the author reported applying focused refinement of different region to improve local resolution of the DMT map. This map was used for interpretation and modeling. This map should be reported, represented and deposited, instead of the 4.1 Å map, which is an intermediate.

Specific and minor comments.

1) Table S1 should be separated into two tables on "Common MIPs" and "Species specific". Since the column "Region" will apply only to the "Common MIPs", not the "Species specific" MIPs.

2) On the structure of FAP115, the authors reported a 3.7Å map. The FSC and the data statistics needs to be reported

3) Figure S3, the caption for C.D panels are missing, please label MT protofilaments

- 4) Page 7 , “These PFs were shown to serve as tracks for anterograde and retrograde intraflagellar transports” please provide reference.
- 5) Figure 4. The authors reported 48nm periodicity for the outer surface filaments at B2/B3, B3/B4, B4/B5, albeit the molecular identity for the outer filament remain unknown, presumably due to low resolution in this region. Please provide evidence how this 48 nm was concluded.
- 6) Fig 5C, please label helices from tubulin and CFAP77 residues
- 7) On Page 17, the correct citation for Relion 4.0 cryoET pipeline should be <https://doi.org/10.1101/2022.02.28.482229>
- 8) Fig 6, the assumption of the extra density being glutamylase is speculative, not substantial. Does the size match the enzyme? Have the authors carried out any focused classification or refinement on this extra density?
- 9) In Fig S2, EF-hand domain (red dashed line) N-terminal domain (black dashed line). This is wrongly labeled. The actual colors are opposite.

Reviewer #2 (Remarks to the Author):

The paper by Kubo, Black, Joachimiak, and colleagues is a high-quality work on the flagellar microtubule doublet (MTD) structure from *Tetrahymena*. Using purified microtubule doublet, they performed cryo-EM and single particle analysis to obtain a structure with a global resolution of 4.1 Å. Then, using artificial intelligent backbone tracing they identified the molecular composition of the different microtubule inner proteins (MIPs). Comparing with the previously published structures of doublet microtubules from *Chlamydomonas* and mammals (bovine), they demonstrate that 50% of the MIPs are conserved but also some MIPs are absent or specific to *Tetrahymena*. The figures are very clear and allow a good understanding of the difference between species and how MTD structures have evolved differently.

A particular point of this article is the identification of proteins at the outer junction between microtubules A and B. The authors identify here CFAP77, a protein conserved in other ciliated species, and test its function by depleting it in *tetrahymena*. As anticipated by the structural localization of CFAP77, the loss of this protein leads to a partial loss of the B microtubule and defects in flagellar movement. It also appears that the loss of CFAP77 affects post-translational modifications on tubulins such as polyglutamylation or polyglutamylation.

Overall, the work presented is of high quality, the text is well-written, the figures are very clear and the results are convincing. I therefore highly recommend the publication of this article, pending minor revisions:

- Figure 2 – the orientation of the view is not clear. In panel A, it is the inner junction, but the section below shows the inner surface of the A-microtubule that corresponds to panel B. It is confusing.

- The title of Figure 5 is mainly about CFAP77, while the outer stabilization has been tested with CFAP77 and OJ2 proteins. Did you run a molecular dynamics simulation in absence of OJ2? Can you quantify the contribution of CFAP77 and OJ2 and the stabilization of the outer junction? If OJ2 is also important to stabilize the outer junction, the title should be rephrased.

- Figure 6: the authors wrote “we occasionally observed gaps in outer junction regions ». Can you quantify this? how many gaps per micron/nm? Are they long gaps? These values might help to better understand the role of CFPA77.

- Figure 6: it would be useful to see a quantification of the length of the flagella to demonstrate that the defect does not affect the assembly but only the stability. If there are gaps on the B-MT, there should also be transport problems (IFT).

- The authors observe an increase in polyglutamylation. Can this value be normalized to the total tubulin signal to verify that there is really more glutamylation and not more flagella? I ask this because I have the impression that there are more flagella images in immunofluorescence

- Figure 6 panel C – A signal of polyglycylated tubulin is visible in the low mag for CFAP77B-3HA but not on the inset. I'm not sure why the signal is lower. Or are the green and red channels inversed?

- Figure 6E-F: In the cryo-tomogram, the B microtubule is missing in some MTD with unknown densities. Could these unknown densities be tubulins? In your model CFAP77 also interacts with the B2 protofilament. Would it be possible to have remaining tubulins of the B1 protofilament? Moreover, it seems that they are present with a periodicity, is this the case?

Reviewer #3 (Remarks to the Author):

The authors used Mass Spectrometry-based proteomics strategies to determine the presence and/or absence of important proteins investigated in this study, for example TtCFAP115, TtRIB22 and TtRIB27 in wild type, RIB72B-KO and RIB72A/B-KO mutants. Indeed, the MS results showed strong evidence for their presence in both wild type and RIB72B-KO mutants given the similar spectral abundance, and for

their absence in the RIB72A/B double knockout mutant given the zero observation. The authors thus further deduced about their interactions with TtRIB72 based on the MS results.

Generally speaking, when a protein is not identified in a conventional shotgun experiment using DDA, one cannot conclude that the protein is missing from the sample, or even below the limit of detection because of the stochastic sampling of DDA. The authors were aware of this fact and thus combined high resolution reconstructions and AI prediction to validate the hypothetical interactions. Another common way for validation is through targeted MS, but given that the spectral abundance in this study is far above the background noise level, further validation might not be necessary.

Overall, the MS experiment design, data analysis approach and interpretation in this study meet the expected standards in the field. The reported MS results are sufficient to support the authors' conclusions.

Reviewer #4 (Remarks to the Author):

The manuscript by Kubo et al describe a cryo-EM model of the thermophila native doublet microtubule. A key strength of the study is depiction of the arrangements of the interior proteins, 38 of which have been assigned. The architectural details of the structure is well explained. However, the way presented this manuscript feels really niche and has little to offer to a broader audience.

For example in the 'DMT consists of conserved and non-conserved MIPs' little is mentioned about the biological relevance of this finding. Periodicity is nothing new in biological assemblies, so it is difficult to comprehend why are the current findings unique. Might be the sheer complexity of 38 interacting subunits that are periodically repeating have some novelty to offer. However, very little is mentioned about the nature (and specificity) of backbone and side-chain interactions (or residue-residue contacts) that manifests in the assembly of the DMT. This analysis can also open up a bioinformatic route of investigation on conservation and co-evolution of key residues, that is missing.

Normally, CG simulations can be employed to extract some sort of elastic properties of the system. But no such attempt has been made.

On a technical side, DeepTracer predictions deteriorate beyond 4 Angstrom. I would suggest some additional validation using molecular dynamics flexible fitting tools like cascade MDFF (suitable for this resolution).

Then the refined geometries should be established with Q-scores and EM-Ringer scores.

Overall, the combination of methods is appealing. However, the manuscript should be reworked to bring forth some take home messages for the general community

First, I would like to thank all the Reviewers for their constructive comments. We have revised the paper significantly in order to address concerns of the Reviewers. In summary, here is what we included in the revised version:

- We updated our methodology for MIP identification by using the *findmysequence* tool. This allowed us to identify 4 more proteins including two PG-rich proteins which bind both inside and outside the doublet microtubules.
- For validation, we used a combination of *findmysequence*, and *in situ* crosslink mass spectrometry to show the accuracy of our identification. In addition, our mass spectrometry data of the RIB72A/B-knockout mutant reflects the accuracy of our MIP identifications by identifying MIPs interacting with RIB72A & B. For the crosslink mass spectrometry, we include data from Dr. Marcotte's laboratory and therefore include them now in the author list.
- Using the validation from *findmysequence* allowed us to observe paralogs of MIPs existing in the doublet microtubule; therefore, we included a supplementary figure highlighting MIP polymorphisms.
- With the identification of the PG-rich proteins and their structures, we were able to show the PG-rich motif is a microtubule-binding motif that can be generalized to the Outer Dense Fiber 3 proteins.
- Updated the MD simulation to simulate the outer junction with and without OJ2 to clear up the role of CFAP77 alone in the stability of the outer junction.
- We reorganized the manuscript and addressed many minor reviewer requests by and also updated our writing to improve clarity and interpretation. The major changes in writing is underscored in our manuscript for easy review.

Since our maps and models are big (431 protein chains), the validation still failed after successful depositions of all our map and models. We had contacted the PDB team for this matter. To make sure about the transparency of our data, we have shared all the deposited map & models here:

<https://www.dropbox.com/sh/ed3wk2xpjgd8o1k/AABgs1fLHbzQ4cfPDMGw7E1Wa?dl=0>

In addition, we also have the real space refine log file from Phenix for our structure as part of the validation.

Below are our point-by-point answers to the reviewers.

REVIEWER COMMENTS

Reviewer #1 (Remarks to the Author):

In this manuscript, Kubo et al. describe a study on the axoneme in *Tetrahymena*. By using cryoEM and single particle analysis on isolated doublet microtubules (DMT), the authors obtained a structure of DMT at 4.1 angstrom. With the aid of protein structure prediction and modeling, this led to atomic models of 38 MIPs found in the map, including 11 MIPs that may not have been previously identified. In addition, the authors described several filamentous structures decorating on the outer surface of DMT, whose molecular identities remain unknown. Finally, the authors characterized the function of two MIPs at outer junction, CFAP77A/B. By analyzing the ciliary beating phenotype of CFAP77A/B knockout and by comparing its structure to wildtype by cryoET, they found the mutant has slower beating rate than the WT. Overall, the techniques the authors had applied are sound. The article is well written, but the organization is a bit disjointed. While the paper appears to focus on the inner junction, there is additional work on other topics added to the manuscript. For instance, analysis of the rib72A/rib72B mutant strain has been published previously and it is unclear what is added in this study. Furthermore, there are several questions concerning the validation of the results and the conclusions presented by the authors.

We thank reviewer #1 for the constructive suggestion. While there is available Rib72A/B mass spectrometry, the MS was done using different methods, as a result, showing different sensitivity and not identical results. We believe that by including our own Rib72A/B mass spectrometry and associated structures, the readers can see the Rib72A/B mass spectrometry as a reliable validation for our identification of those MIPs (top 6 missing proteins identified as MIPs), whose assembly requires RIB72A or RIB72B. Following reviewer #1's suggestion, we reorganized the figures and text to make the writing more coherent.

Major comments.

1) The authors claimed to have identified 11 new MIPs that are unique to *Tetrahymena*. However, the list of MIPs presented here are not well rectified with previous studies of the *Tetrahymena* axoneme. There is some discussion of the topic, but it is difficult to easily identify newly identified MIPs versus previously known MIPs. Perhaps the MIPs listed in Tables S1 and S2 could be annotated to identify previous studies in which the given MIP was identified.

We changed Supplementary Tables 1 and 2 as suggested from Reviewer #1.

2) The authors named non-conserved “auxiliary” MIPs, while the authors classified the rest of MIPs that have been found in other species as “core” MIPs. From an evolutionary point of view, perhaps this is correct, but this is likely not true as these MIPs’ functions concern. It’s likely that the auxiliary MIPs will be equally important for the cilia stability or waveform regulation. Until further characterizing the function of these MIPs, calling them “core” or “auxiliary” could be misleading about function and other names for the groups should be considered.

This is an excellent point. We now use “Conserved” and “Species-specific” in the manuscript. We also included the point about the perhaps “equal importance” of “auxiliary MIPs” as per the following text:

“To summarize, each DMT region has core components while other MIPs act as species-specific members, possibly to reinforce the function of the core component, appropriate for specific types of motilities. As such, the species-specific MIPs can also be essential for cilia stability and waveform regulation.”

3) The authors estimated the overall resolution of the 48nm long DMT at 4.1 angstrom. Furthermore, by using focused refinement, the local resolution has been improved to 3.6 to 3.9 Å. The authors need to provide a local resolution estimate for validation.

We updated Supplementary Figure 1 to include our local resolution/focus refinement strategy, as well as K40R and 96-nm map FSC curves.

4) Concerning identifying and model building of the MIPs, the authors have adapted two parallel approaches and have provided example for each in Figure S1B. In both examples, the MIPs (identities not indicated in Fig S1B) have a globular domain. Perhaps these two represent the best scenario. It is well known that tracing protein backbone and deciding the direction of the chain are difficult when the target structure is most coiled-coil or unstructured loop. This is exacerbated when the map resolution is lower than 4 Å where the most side-chains are invisible. The authors need to provide detailed examples on these MIPs, for example, RIB38, NIP21A and MIP26.

This is a valid point from Reviewer #1. In our experience, tracing backbone, deciding direction, and determining identity is very difficult with coiled coil proteins which tend to not bind tightly to the tubulin lattice. Intrinsically disordered MIPs can be, in fact, easier identified using the *findmysequence* approach due to the fact that disordered proteins tend to bind very tightly to the tubulin lattice (therefore, good resolution) and many bulky side chains are available to complement the negative charge of tubulin (See Supplementary Table 4 for the E-value of disorder protein such as SPTG1, SPTG2 or

SB1, CFAP182A/B, CFAP129). Fortunately for us, most of the hard to identify coiled coil proteins (filamentous MIPs) are already identified in other studies such as CFAP53, CFAP45, CFAP210 and CFAP127.

Regarding resolution for identification, we identified many MIPs using the K40R map, which has local resolutions between 3.2 - 3.5 Angstrom (see accompanying data: *k40r_composite.ccp4* or *k40r_composite_deepenhancer.mrc*). We made sure to look at the same density in the CU428 (*cu428_composite.ccp4* and *cu428_composite_deepenhancer.mrc*) for validation as well. We provide an updated Supplementary Figure 2 with examples of each MIP structure and its density.

Supplementary Table 4 shows the E-value of 1st, 2nd and 3rd matches. It shows that the 1st match is most of the time significantly better than the second match. In these cases, the matches are not as clear, the second match is always the paralog with very similar sequence identity (such as CFAP77A & B).

In addition, we also show the *in situ* crosslink mass spectrometry in Supplementary Figure 2 and Supplementary Table 5 to show that MIPs have reliable crosslinks to the luminal side of tubulin. In our experience, most of the time the direction of the chain is estimated accurately by Deept racer even at 4 Angstrom resolution. In rare cases, the direction is wrong or not clear; in those cases, we actually trace the backbone in both directions and then use *findmysequence* for identification. The result is very clear on which direction is correct.

5) Interestingly, the authors have found the FAP115 having a periodicity of 32 nm. This is consistent with the size TtFAP115 that is about 4 times larger than CrFAP115. However, since the 4 EF domains in FAP115 are not identical, how do authors know which map density is attributed to domain 1,2,3,4 respectively? Fig S3A showed subtle difference among 4 EF-hand domains, but no detail has been provided. As authors correctly pointed out that the unique 32nm periodicity of FAP115 implies that entire DMT having a periodicity of 96 nm. Meanwhile, the location of all MIPs in DMT are coherent, it will be critical to understand the position of FAP115 molecular and its domains relative to other MIPs, for example MIPs with 48 nm periodicity. The authors have not provided this information.

We have updated Supplementary Figure 3I to show the distinction between each of the 4 EF-hand domain pairs of TtCFAP115. Interestingly, when we compared the 96-nm map & 48-nm map, it looks almost identical. Therefore, the 32-nm repeat of TtFAP115 does not interfere too much with other MIPs, perhaps, due to the similarity of the four domains.

6) The authors found several fMIP at the outer surface of DMT. The sites coincide with the binding sites for kinesin or dynein IFT trains. The authors draw comparison of these fMIPs to MAP7 that reportedly has a biphasic regulatory activity for kinesin-1. However, in this case the kinesin-1 activity increases only when the MAP7 concentration is low and when there are many available sites on MT unoccupied by MAP7. Given that the outer surface of DMT is fully occupied by fMIP, the mechanism of IFT trimeric kinesin-2 binding on that surface will likely be different from that of MAP7 decorated MT. The authors' hypothetical analogy is questionable.

The fMIPs on the outer surface are not fully occupied since their densities are significantly weaker than the microtubule. As a result, the resolution of the densities of the fMIPs on the outer surface is much lower, perhaps due to low occupancy. We added the following text to to clarify that point:

“The low resolution of the outer surface filament suggests that they are either partially decorated or inherently flexible.”

“High concentration of MAP7 seems to inhibit kinesin-1 activity (Ferro et al., 2022). Therefore, it is reasonable to suggest that the low resolution of the outer surface filament is since it is partially decorated and, therefore, not inhibitory to kinesin-2 activities.”

7) The authors studied the CFAP77A/B KO by cryoET and subtomogram averaging, and had observed no difference compared to the WT. Fig 6D shows the two averages but lacking any labels that depict their orientations or indicate the colored densities. Is the CFAP77A/B density missing the double KO average?

We moved the 96-nm maps of CFAP77A/B-KO and WT to Supplementary Figure 7. We labeled them properly as well. Unfortunately, we cannot see the missing CFAP77 due to the lack of resolution using cryo-ET and subtomogram averaging. Perhaps, we can only detect the missing CFAP77 at ~5-8 Angstrom resolution.

The authors reported the subtomogram averaged at 18 and 21 angstroms for the WT and the mutant. Without reporting their FSCs or the number of subtomogram used, it is difficult to judge the quality of the results.

We added the subtomogram averaged FSC curves to Supplementary Figure 7. The number of subtomograms used is now updated in Table 1 and Materials & Methods.

In Fig 6E and F, the authors reported DMT defects in the raw tomograms, have they tried 3D classification to identify these defects in subtomogram average? Finally, 0.15% glutaraldehyde was used for sample preparation. It is well-known this non-specific cross-linking reagent is prone to introduce artefact. This raises concern on the interpretation of the result from cryoET.

We tried 3D classification but no “defect class” was detected, perhaps due to the irregular occurrence of defects.

0.15% glutaraldehyde is the concentration used in GraFix methods, which allows for protein complexes to remain intact but minimizes aggregation and artifacts. In fact, previous studies such as Zhang et al., 2017, Cell 169:1303-1314 have shown that you can still obtain high resolution with 0.15% glutaraldehyde. In this case, we want to preserve the round shape of the cilia be able to interpret the data appropriately. In our experience, the crosslinked WT cilia show no difference from non-crosslinked WT and membrane WT cilia. We include the following sentence to the *Cryo-ET acquisition and reconstruction* section in **Materials & Methods** to address that:

“The crosslinked axonemes appear similar to non-crosslinked axoneme without any obvious artifacts.”

8) In “Material and Methods”, the author reported applying focused refinement of different region to improve local resolution of the DMT map. This map was used for interpretation and modeling. This map should be reported, represented and deposited, instead of the 4.1 Å map, which is an intermediate.

To increase the usability of our map, we used the *phenix.combine_focus_map* function to create a composite map, which picks the highest resolutions from the different focused maps. This single composite map allows for the visualization of the doublet microtubule at a rather uniform resolution instead of loading more than 20 different focused refined maps. We are depositing this map from *WT* and *K40R* as well as the 96-nm map and the subtomogram averaged map.

Specific and minor comments.

1) Table S1 should be separated into two tables on “Common MIPs” and “Species specific”. Since the column “Region” will apply only to the “Common MIPs”, not the “Species specific” MIPs.

We split and updated the original Supplementary Table 1 into Supplementary Tables 1 and 2.

2) On the structure of FAP115, the authors reported a 3.7A map. The FSC and the data statistics needs to be reported

We added this information to Tables 1 & 2 and included the FSC curves in Supplementary Figure 1.

3) Figure S3, the caption for C.D panels are missing, please label MT protofilaments

We actually removed those panels to improve the flow of the paper. Supplementary Figure 3 now is mainly about MIP paralogs.

4) Page 7, “These PFs were shown to serve as tracks for anterograde and retrograde intraflagellar transports” please provide reference.

Citation updated.

5) Figure 4. The authors reported 48nm periodicity for the outer surface filaments at B2/B3, B3/B4, B4/B5, albeit the molecular identity for the outer filament remain unknown, presumably due to low resolution in this region. Please provide evidence how this 48 nm was concluded.

It is true that 48-nm is imposed on the 48-nm map. In fact, we observe the same densities on the 96-nm so we believe the 48-nm periodicity is correct for those filaments. We updated the text to clear that up:

“At this resolution, the outer surface filaments of PFs A9A10, A10B1 and B1B2 still exhibit a clear 24-nm periodicity (Fig. 4B). Other filaments might have 48-nm repeat since they appear similarly in the 96-nm map.”

6) Fig 5C, please label helices from tubulin and CFAP77 residues

Helix label added.

7) On Page 17, the correct citation for Relion 4.0 cryoET pipeline should be <https://doi.org/10.1101/2022.02.28.482229>

Citation added.

8) Fig 6, the assumption of the extra density being glutamylase is speculative, not substantial. Does the size match the enzyme? Have the authors carried out any focused classification or refinement on this extra density?

It is true that it is entirely speculative due to the increase in polyglutamylated. The density is not regular so it does not show in the average, unfortunately. We updated the text to reflect that information:

“Based on the increase in polyglutamylated in the *CFAP77A/B-KO* mutants and the presence of proteins outside the DMT, we speculate that the destabilization of the DMT might lead to a compensatory effect in post-translational modification. As the result, the polyglutamylated increase in the DMT might compensate for the de-stabilization from the lack of CFAP77.”

9) In Fig S2, EF-hand domain (red dashed line) N-terminal domain (black dashed line). This is wrongly labeled. The actual colors are opposite.

We changed Supplementary Figure 2 and that is fixed.

Reviewer #2 (Remarks to the Author):

The paper by Kubo, Black, Joachimiak, and colleagues is a high-quality work on the flagellar microtubule doublet (MTD) structure from *Tetrahymena*. Using purified microtubule doublet, they performed cryo-EM and single particle analysis to obtain a structure with a global resolution of 4.1 Å. Then, using artificial intelligent backbone tracing they identified the molecular composition of the different microtubule inner proteins (MIPs). Comparing with the previously published structures of doublet microtubules from *Chlamydomonas* and mammals (bovine), they demonstrate that 50% of the MIPs are conserved but also some MIPs are absent or specific to *Tetrahymena*. The figures are very clear and allow a good understanding of the difference between species and how MTD structures have evolved differently.

A particular point of this article is the identification of proteins at the outer junction between microtubules A and B. The authors identify here CFAP77, a protein conserved in other ciliated species, and test its function by depleting it in *tetrahymena*. As anticipated by the structural localization of CFAP77, the loss of this protein leads to a partial loss of the B

microtubule and defects in flagellar movement. It also appears that the loss of CFAP77 affects post-translational modifications on tubulins such as polyglycylation or polyglutamylaton.

Overall, the work presented is of high quality, the text is well-written, the figures are very clear and the results are convincing. I therefore highly recommend the publication of this article, pending minor revisions:

- Figure 2 – the orientation of the view is not clear. In panel A, it is the inner junction, but the section below shows the inner surface of the A-microtubule that corresponds to panel B. It is confusing.

We changed Figure 2 to make it clear as suggested.

The title of Figure 5 is mainly about CFAP77, while the outer stabilization has been tested with CFAP77 and OJ2 proteins. Did you run a molecular dynamics simulation in absence of OJ2? Can you quantify the contribution of CFAP77 and OJ2 and the stabilization of the outer junction? If OJ2 is also important to stabilize the outer junction, the title should be rephrased.

Thanks for the suggestion, we did MD in the absence of OJ2 (Supplementary Fig. 5E) and the following table. It is clear that CFAP77 is more important for the stability of the outer junction.

Average energy between A- and B-tubule

	With OJ2	Without OJ2
With CFAP77	-6.2604 ± 0.00408215	-6.05342 ± 0.0055639
Without CFAP77	-7.18253 ± 0.00409181	-0.290828 ± 0.00524678

To see this result, we're sure that CFAP77 mainly keep A-B stability because the average bounding energy with and without OJ2 case is almost same. Interestingly, however, only the OJ2 case (without CFAP77) looks more stable than the WT case. So, we add additional analysis.

Average energy between B1 and B2

	With OJ2	Without OJ2
With CFAP77	-30.2775 ± 0.00598608	-30.7353 ± 0.00608574
Without CFAP77	-27.5101 ± 0.00881373	-26.6082 ± 0.00922062

To see this result, we're sure that both exist case and only with CFAP77 case is almost same. Besides, the energy between B1 and B2 in the only OJ2 exist case changes weaker than WT case.

From these results, we can say that CFAP77 is the key MIPs for building stable B-tubule along the A-tubule.

We updated the following in the text to reflect that.

“Molecular dynamics simulations indicate that mainly TtCFAP77 contributes to A- and B-tubule stability because the energy with and without OJ2 is almost the same (Supplementary Fig. 5E). In addition, further analysis shows that B-tubule bounding angle with A-tubule become unstable in the absence of both TtCFAP77 and OJ2 (Supplementary Fig. 5F). Therefore, TtCFAP77 is the key MIP for the stable binding of the B-tubule to the A-tubule.”

- Figure 6: the authors wrote “we occasionally observed gaps in outer junction regions ». Can you quantify this? how many gaps per micron/nm? Are they long gaps? These values might help to better understand the role of CFPA77.

Unfortunately, the gap is not very frequent and our tomograms contain short lengths of cilia (< 1 micron). Quite a few CFAP77A/B-KO tomograms show no gap at all. However, with our experience working with WT tomograms, we don't normally see this defect. We reflected on this in the following text:

“However, when we examined the raw tomograms of the *CFAP77A/B-KO* mutant, we observed gaps in outer junction regions in a few tomograms but not all. This suggests that missing CFAP77 likely destabilizes the outer junction mildly. We also observed some unknown and non-periodic densities binding to the outer junction from outside in some cilia in the *CFAP77A/B-KO* mutant (Fig. 6E, F, red arrows).

- Figure 6: it would be useful to see a quantification of the length of the flagella to demonstrate that the defect does not affect the assembly but only the stability. If there are gaps on the B-MT, there should also be transport problems (IFT).

We added the cilia length quantification in Figure 6. And added the following text:

“The cilia length of two clones of *CFAP77A/B-KO* is about 90% wild type cells suggesting mild assembly defects (Fig. 6C).”

and added details in Fig 6C description.

”On average cilia length was: WT=5.85 μm (number of measured cilia, n=115), *CFAP77A/B-KO* clone 1=5.46 μm (n=74), *CFAP77A/B-KO* clone 2=5.27 μm (n=83). Student t-test WT/KO is 2E-08 and 6,8E-16, respectively.”

- The authors observe an increase in polyglutamylation. Can this value be normalized to the total tubulin signal to verify that there is really more glutamylation and not more flagella? I ask this because I have the impression that there are more flagella images in immunofluorescence

The *CFAP77A/B-KO* mutants assemble a similar number of cilia as WT cells. We assume that the impression of more numerous cilia in the mutant is because of the stronger ciliary signal on the presented image (due to a higher level of glutamylation detected by polyE) while cilia in WT cells stained side-by-side with *CFAP77A/B-KO* cells are poorly visible. Importantly, the measurements of the level of tubulin glutamylation in cilia in WT and *CFAP77A/B-KO* cells fixed side-by-side done using ImageJ program, clearly showed that the level of this posttranslational modification is, on average, elevated in the mutant cilia.

Similarly, the elevated level of glutamylation in cilia assembled by *CFAP77A/B-KO* mutant was revealed using western blotting. Moreover, in this case, we not only showed the increase in the number of long polyE side chains (anti-polyE antibody) but also an increase in the total number of the glutamyl side chains (GT335 antibody). The level of tubulin glutamylation detected by either polyE or GT335 antibodies was normalized to the level of tubulin using anti-alpha tubulin 12G10 antibodies (we measured the intensity of both anti-12G10 and anti-GT333 or polyE positive bands) and the graph presented as Supplementary Figure 7F shows normalized values (the ratio of the glutamylated tubulin to α -tubulin).

- Figure 6 panel C – A signal of polyglycylated tubulin is visible in the low mag for CFAP77B-3HA but not on the inset. I'm not sure why the signal is lower. Or are the green and red channels inverted?

We apologize for the confusion. By mistake, the CFAP77B-3HA inset shows cilia stained with anti-HA antibody (only signal from one channel) instead of the merged image. In the revised version of the manuscript, we corrected this mistake. Additionally, we added insets of the same area, but showing signals from both channels slightly shifted, so the green signal (anti-HA) is parallel (instead of overlapping) with a red signal (polyG, polyglycylated tubulin). Please note that such a shift revealed a very weak signal of CFAP77A-3HA in the distal half of the cilia (on this merged image, the green signal was strongly enhanced).

- Figure 6E-F: In the cryo-tomogram, the B microtubule is missing in some MTD with unknown densities. Could these unknown densities be tubulins? In your model CFAP77 also interacts with the B2 protofilament. Would it be possible to have remaining tubulins of the B1 protofilament? Moreover, it seems that they are present with a periodicity, is this the case?

We don't know the identity of the unknown density. It could be tubulin. However, it doesn't look regular. As a result, it does not show up in the subtomogram average. Where we observed the unknown density, the B-microtubule seems to be intact. Figure 6E & F are from different tomograms.

Reviewer #3 (Remarks to the Author):

The authors used Mass Spectrometry-based proteomics strategies to determine the presence and/or absence of important proteins investigated in this study, for example TtCFAP115, TtRIB22 and TtRIB27 in wild type, RIB72B-KO and RIB72A/B-KO mutants. Indeed, the MS results showed strong evidence for their presence in both wild type and RIB72B-KO mutants given the similar spectral abundance, and for their absence in the RIB72A/B double knockout mutant given the zero observation. The authors thus further deduced about their interactions with TtRIB72 based on the MS results.

Generally speaking, when a protein is not identified in a conventional shotgun experiment using DDA, one cannot conclude that the protein is missing from the sample, or even below the limit of detection because of the stochastic sampling of DDA. The authors were

aware of this fact and thus combined high resolution reconstructions and AI prediction to validate the hypothetical interactions. Another common way for validation is through targeted MS, but given that the spectral abundance in this study is far above the background noise level, further validation might not be necessary.

Overall, the MS experiment design, data analysis approach and interpretation in this study meet the expected standards in the field. The reported MS results are sufficient to support the authors' conclusions.

The Reviewer is totally right about the stochastic nature of MS. We believed that the combination of MS and high-resolution structures gives a good validation of our MIP identification.

Reviewer #4 (Remarks to the Author):

The manuscript by Kubo et al describe a cryo-EM model of the thermophila native doublet microtubule. A key strength of the study is depiction of the arrangements of the interior proteins, 38 of which have been assigned. The architectural details of the structure is well explained. However, the way presented this manuscript feels really niche and has little to offer to a broader audience.

For example in the 'DMT consists of conserved and non-conserved MIPs' little is mentioned about the biological relevance of this finding. Periodicity is nothing new in biological assemblies, so it is difficult to comprehend why are the current findings unique. Might be the sheer complexity of 38 interacting subunits that are periodically repeating have some novelty to offer. However, very little is mentioned about the nature (and specificity) of backbone and side-chain interactions (or residue-residue contacts) that manifests in the assembly of the DMT. This analysis can also open up a bioinformatic route of investigation on conservation and co-evolution of key residues, that is missing.

We thank the reviewer for the helpful comment. In fact, we explicitly tried to not go deep into interactions, which would make the manuscript a bit more disjointed because there are many things that can be learned about the interactions between MIPs and tubulin. And I believe that there is a manuscript out there (not from our group) on this specific aspect from a bioinformatic angle.

To partly address this comment, we updated Supplementary Figure 3 to include the MIP paralogs, which perhaps inspires further work on comparisons across paralogs and species for the backbone and side-chain interactions. In addition, we present the residue-residue contact from the PG-motif in our newly identified proteins, which allows us the

understanding of outer dense fiber 3 protein to microtubules in Figure 4 and Supplementary Figure 4.

Normally, CG simulations can be employed to extract some sort of elastic properties of the system. But no such attempt has been made.

To extract the elastic properties of the entire flagellar system, we need to simulate an entire doublet microtubule and more. However, it is not possible due to the time cost (a too big system consisting of many proteins for our coarse-grained level; in which one amino-acid is one bead). Alternatively, an elastic network model or rougher CG model, likely one protein complex is one bead, can be used on the entire cilia or doublet microtubule, but in that case, we can't get atomistic-level information on OJ2 and CFAP77 as we did.

We don't show extract the elastic properties, but we additionally try to calculate angular elasticity for A- and B-tubules bounding from our MD simulations. Our MD system included three pairs of alpha-beta tubulin dimers along the direction of microtubule elongation. The two vectors used in the angle calculation were defined by the K401 of the central alpha-tubulin in A10 as a vector base point, and the center of mass of the central alpha-tubulin in A12 and B2 as an endpoint of the vector. Since K401 is a residue anchored in space for MD simulation efficiency, it serves as a base point for following the change in the angle between A and B tubules.

The results are shown in the figure: WT and without both CFAP77 and OJ2 cases. The perpendicular line in the figure is the initial state angle. Since the simulations were originally performed by cutting out only a portion of the ring-shaped doublet microtubule, it is easy to see that the angles for both cases often show smaller than the crystal structure. However, it is clear that the angle fluctuates more greatly in the absence of CFAP77 and OJ2 cases. This result is consistent with our previous result that the absence of CFAP77 and OJ2 destabilizes the binding energy of A-B tubules.

We updated the following in the text to reflect that.

“Also, Molecular dynamic shows that B-tubule bounding angle with A-tubule become unstable in the absence of both TtCFAP77 and OJ2 (Supplementary Fig.5F).”

On a technical side, DeepTracer predictions deteriorate beyond 4 Angstrom. I would suggest some additional validation using molecular dynamics flexible fitting tools like cascade MDFF (suitable for this resolution).

Then the refined geometries should be established with Q-scores and EM-Ringer scores.

Thank you for the suggestion. As mentioned above, we have improved the writing/visualization of validation.

Regarding resolution for identification, we identified many MIPs using the K40R map, which has local resolutions between 3.2 - 3.5 Angstrom (see accompanying data: k40r_composite.ccp4 or k40r_composite_deepenhancer). Even our WT's local resolution is between 3.6 - 3.9 Angstrom. We made sure to look at the same density in

the CU428 for validation as well. We provide a new Supplementary Figure 2 with examples of each MIP structure and its density.

Supplementary Table 4 shows the E-value of 1st, 2nd and 3rd matches. It shows that the 1st match is significantly better than the second match, except for close paralogs. In cases where the matches are not as clear, the second match is always the paralog with a very similar sequence identity (such as CFAP77A & B).

In addition, we also show the *in situ* crosslink mass spectrometry in Supplementary Figure 2 and Supplementary Table 5 to show that MIPs have reliable crosslinks to the luminal side of tubulin. In our experience, most of the time the direction of the chain is estimated accurately by Deeptracer even at 4 Angstrom resolution. In very rare cases, the direction is wrong or not clear; in those cases we actually trace the backbone in both directions and then use *findmysequence* for identification. The result is very clear on which direction is correct.

Overall, the combination of methods is appealing. However, the manuscript should be reworked to bring forth some take home messages for the general community

Thanks for the suggestion. We have changed a lot of the figures and writing in our manuscript to improve the clarity and the take-home message such as the PG motif and the periodicity determination of the outside and inside of the doublet. The newly added parts are under-scored in the revised manuscript.

REVIEWERS' COMMENTS

Reviewer #1 (Remarks to the Author):

In this revised version, Kubo et al. have substantially improved the clarity of the manuscript. They presented additional data supporting their original conclusions. Furthermore, the authors have identified 4 new proteins with multiple PG-rich motifs, enriching a growing list of proteins associated to DMT.

The authors have addressed all questions and concerns previously raised by this reviewer. The study is comprehensive and the manuscript is in excellent shape for publication. Kudos to the authors on this study.

A number of minor typographical errors:

Page 8, in several places, "STPG1A and STPG2" are mis-spelled as "SPTG1A and SPTG2"

Page 8, "which is believed to the site of" should be "which is believed to be the site of"

Page 10, "Coincidentally, more TtCFAP77 is present in the proximal half of the cilium (Fig. 6C); possibly the B-tubule is more stable in that region." should be "Coincidentally, more TtCFAP77A is present in the proximal half of the cilium (Fig. 6D); possibly the B-tubule is more stable in that region."

Reviewer #3 (Remarks to the Author):

The authors included additional crosslink mass spectrometry data to demonstrate the interactions between MIPs and the luminal side of tubulin. The results shown in Supplementary Figure 2 and Supplementary Table 5 are sound. The detailed description of the crosslinking study in Marcotte's work including sample preparation, MS analysis and data interpretation meets the expected standards in the field. I do not have further comments.

Reviewer #4 (Remarks to the Author):

My comments are now addressed.